# CSOR: Coreset Selection for Object Re-identification via Class Pruning

Minyoung Oh [1]    Jae-Young Sim [1]

## Abstract

Coreset Selection (CS) aims to extract a small yet representative subset from a large dataset, reducing the complexity of model training. Although CS has been primarily investigated for classification tasks, it is still underexplored for object Re-identification (ReID). In this paper, we first formulate Coreset Selection for Object Re-identification (CSOR) as a joint optimization problem to find both the optimal coreset and the optimal class subset. We identify intra-class diversity as a key factor for effective coreset construction for ReID. Based on this insight, we propose a novel two-stage framework, consisting of Diversity-driven Class Pruning (DCP) and Coverage-Prioritized Sampling (CPS), to address the unique challenges of ReID datasets. First, classes with low feature diversity are pruned to allocate the storage budget to the remaining informative classes. Then, samples are greedily selected in an easy-to-hard class order to maximize feature coverage within each class. Extensive experiments on three person ReID datasets and one vehicle ReID dataset demonstrate that our method consistently outperforms existing CS approaches, establishing a new state-of-the-art in CSOR.

## 1. Introduction

As modern datasets continue to grow in scale, the corresponding demands on storage and training resources have also increased. Coreset Selection (CS) refers to the task of selecting a small yet representative subset from a large dataset, such that models trained on this subset achieve performance comparable to those trained on the entire dataset (Toneva et al., 2019; Zheng et al., 2023). Therefore, CS has attracted considerable attention as a promising

[1]Graduate School of Artificial Intelligence, Ulsan National Institute of Science and Technology (UNIST), Republic of Korea. Correspondence to: Jae-Young Sim <jysim@unist.ac.kr>.

*Proceedings of the 43$^{rd}$ International Conference on Machine Learning*, Seoul, South Korea. PMLR 306, 2026. Copyright 2026 by the author(s).

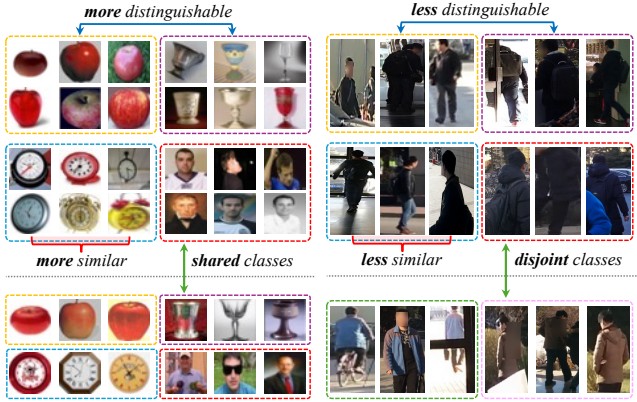

*(a) Classification dataset*   *(b) ReID dataset*

*Figure 1.* Characteristics of datasets used for (a) classification and (b) ReID. The upper and lower rows show the training and testing datasets, respectively. Different colors represent different classes.

approach to reduce memory requirements and training complexity, and it is applicable to various resource-constrained settings, including continual learning (Wang et al., 2024; Shin et al., 2017), federated learning (McMahan et al., 2017; Hao et al., 2025), and neural architecture search (Elsken et al., 2019; Shim et al., 2021).

However, CS for object re-identification (ReID)—the task of retrieving gallery images that match the identity of a query image—remains underexplored, despite ReID being a particularly compelling target for CS. ReID datasets are typically collected from multi-camera video streams, resulting in numerous duplicated frames over time and high temporal redundancy. For example, the MARS (Zheng et al., 2016) dataset, collected across six cameras, contains over one million person images, where consecutive frames within a tracklet are often nearly identical. This redundancy inflates storage and training costs without proportionally enriching the training signal. Motivated by this, we introduce a novel problem: **Coreset Selection for Object Re-identification (CSOR)**. CSOR serves as a foundational task for mitigating the inherent temporal redundancy of ReID datasets, substantially reducing storage and training costs.

We observe that ReID datasets exhibit characteristics that are distinct from those of classification datasets. Figure 1 illustrates the key differences. ***Observation 1:*** In classification datasets, classes typically have distinct visual ap-

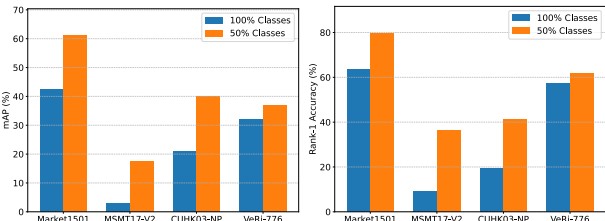

*Figure 2.* Comparison of ReID performance between a coreset uniformly sampled from all classes (100%; blue) and a coreset sampled from a randomly selected half of the classes (50%; orange), under a limited storage budget of 1,000 images.

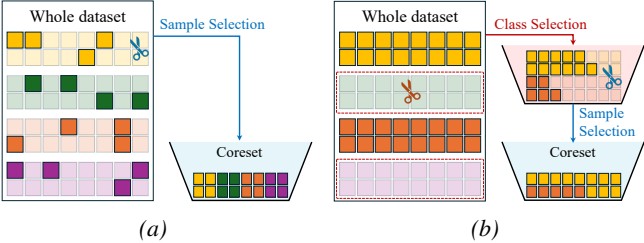

*Figure 3.* Class pruning-based coreset selection for object ReID. (a) The conventional CS approach for classification tasks, which directly selects samples from all classes. (b) The proposed two-stage framework for ReID, in which classes are first pruned, followed by sample selection from the remaining classes.

pearances, making them relatively easy to distinguish. In contrast, classes in ReID datasets correspond to different identities while belonging to the same semantic category (e.g., person or vehicle). As a result, images from different classes often share similar overall appearances, leading to reduced inter-class distinctness. ***Observation 2:*** In classification tasks, the label space is typically shared between the training and test sets. In ReID tasks, however, the identity (label) spaces of the training and test sets are disjoint.

*Observation 1* suggests that preserving sufficient intra-class diversity in ReID datasets is crucial for learning identity-discriminative features. Achieving this typically requires a sufficient number of samples per class; however, under a limited storage budget, it is difficult to retain many images for every class. Importantly, unlike in standard classification settings, it may not be necessary to keep samples from all classes by *Observation 2*. Motivated by this observation, we conduct a simple experiment in which we prune a subset of classes to allocate more samples to the remaining classes. Figure 2 compares ReID performance using two coresets under the same budget: one (blue) uniformly samples images from all classes, whereas the other (orange) uniformly samples images from only a randomly selected half of the classes. With the storage budget fixed to 1,000 images, the latter coreset contains twice as many images per class as the former, potentially capturing richer intra-class diversity. As shown in Figure 2, the coreset with higher intra-class diversity consistently yields better performance across four ReID datasets.

Based on this finding, we formulate the CSOR problem to identify not only an optimal set of samples but also an optimal subset of classes, as illustrated in Figure 3. Specifically, while conventional coreset selection for classification preserves the original label set and selects images from all classes, CSOR first selects informative classes and then chooses representative images from the remaining classes. To remove uninformative classes, we propose *Diversity-driven Class Pruning* (DCP), which discards classes with low intra-class diversity, as they contribute little to learning identity-discriminative features. By doing so, we allocate more budgets to select more images from informa-

tive classes under a fixed storage constraint, thereby improving intra-class diversity. Subsequently, we introduce a *Coverage-Prioritized Sampling* (CPS) method, which prioritizes samples that maximize feature coverage within each class. Based on the facility location function (Owen & Daskin, 1998), which satisfies the property of submodularity, we efficiently maximize feature coverage by iteratively selecting samples in a greedy manner. Extensive experiments have been conducted on three major person ReID datasets and one vehicle ReID dataset. Experimental results demonstrate that the proposed method generates coresets with higher intra-class diversity and substantially outperforms existing CS methods.

Our contributions are summarized as follows:

- To the best of our knowledge, we are the first to introduce the problem of coreset selection for object ReID under a fixed storage budget, whereas conventional CS methods focus on classification tasks.

- We propose a novel two-stage framework that first prunes uninformative classes based on their diversity scores and then adaptively samples images to maximize feature coverage within each class.

- We conduct extensive experiments on four major ReID datasets, demonstrating that our method significantly outperforms all existing CS methods.

## 2. Related Work

### 2.1. Coreset Selection

Coreset selection, also known as data pruning, aims to reduce the training dataset size by selecting informative samples according to predefined criteria (Yang et al., 2024; Pleiss et al., 2020; Zheng et al., 2023; Welling, 2009; Xia et al., 2023; Paul et al., 2021; Toneva et al., 2019). Forgetting (Toneva et al., 2019) counts how many times a sample is misclassified after being correctly classified and considers

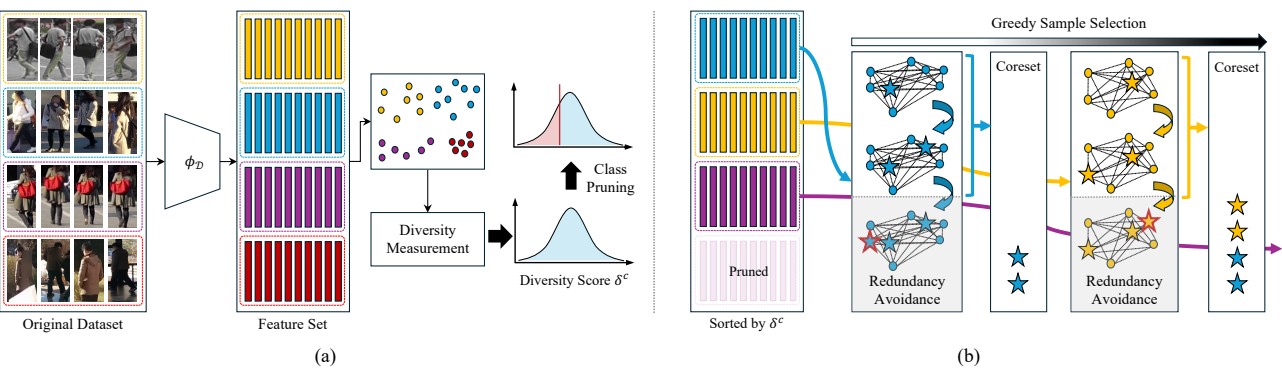

*Figure 4.* Overview of the proposed CSOR framework. (a) Diversity-driven Class Pruning (DCP) quantifies intra-class feature diversity to identify and prune less informative classes. (b) Coverage-Prioritized Sampling (CPS) iteratively selects samples that maximize the marginal gain in feature coverage within each class, explicitly suppressing redundancy among the selected samples.

samples with a high count to be informative. AUM ([Pleiss et al.](#), 2020) computes the average prediction margin of each sample during training and selects those with low scores that are likely to be mislabeled. EL2N ([Paul et al.](#), 2021) calculates the L2 norm of the prediction error vector for each sample at an early stage of training, selecting samples with higher norms. GraNd ([Paul et al.](#), 2021) selects samples based on the norm of the loss gradient with respect to the model's parameters, prioritizing samples with larger gradient norms. Moderate ([Xia et al.](#), 2023) selects samples with intermediate prediction confidence, based on the observation that both very easy and very hard samples are less informative. CCS ([Zheng et al.](#), 2023) selects a subset that maximizes the feature space coverage of the entire dataset, particularly targeting high pruning rate scenarios. Recently, CS methods for various fields such as federated learning ([Hao et al.](#), 2025) and object detection ([Lee et al.](#), 2024) have been proposed, while CS for ReID has not been explored yet. It is necessary to develop a CS method specialized for ReID due to the unique characteristics of ReID datasets, which differ from those of classification datasets.

### 2.2. Object Re-identification

Object Re-identification (ReID) aims to retrieve images of the same identity as a given query image from a gallery set. It has been widely studied in various domains, especially in person ReID ([Luo et al.](#), 2019; [He et al.](#), 2021) and vehicle ReID ([Liu et al.](#), 2016) due to their wide applicability. ReID frameworks are typically based on contrastive learning, such as the triplet loss ([Schroff et al.](#), 2015), to learn fine-grained feature representations. This approach learns to pull together features of the same identity while pushing apart those of different identities. For effective contrastive learning, datasets are known to require a sufficient number of positive and negative samples of varying difficulty ([Robinson et al.](#), 2021; [Chen et al.](#), 2020). However, existing CS methods are not well-suited for ReID, as

they were primarily developed for classification tasks and typically select informative samples based on dataset-level criteria, often ignoring class-wise characteristics. Moreover, most methods assume that sample difficulty correlates with informativeness, an assumption that is detrimental in ReID, where difficult samples frequently correspond to noisy outliers rather than informative examples, thereby impeding the learning of discriminative features. To address these limitations, we propose a novel CS framework specifically designed for ReID, which prioritizes intra-class diversity.

## 3. Methodology

### 3.1. Problem Formulation

Let $\mathcal{D} = \bigcup_{c \in \mathcal{C}} \mathcal{D}_c$ be a training dataset composed of class-wise subsets $\mathcal{D}_c = \{(\boldsymbol{x}, y) \mid y = c\}$, where $\mathcal{D}_c$ denotes the set of samples belonging to class $c \in \mathcal{C}$, and $\boldsymbol{x}$ and $y$ represent a sample image and its label, respectively. The goal of coreset selection is to construct a subset $\hat{\mathcal{D}} \subset \mathcal{D}$, referred to as a coreset, with $|\hat{\mathcal{D}}| \leq B \ll |\mathcal{D}|$, under a fixed storage budget $B$, such that a model $\phi_{\hat{\mathcal{D}}}$ trained on $\hat{\mathcal{D}}$ achieves performance comparable to that of a model $\phi_{\mathcal{D}}$ trained on the full dataset $\mathcal{D}$.

**Coreset Selection for Classification** In classification tasks, the optimal coreset $\hat{\mathcal{D}}^*$ can be obtained as:

$$\hat{\mathcal{D}}^* = \arg \min_{\hat{\mathcal{D}} \subset \mathcal{D}} \mathbb{E}_{\boldsymbol{x} \sim P} \big[ \mathcal{L}_{\text{CE}}(\boldsymbol{x}, y; \phi_{\hat{\mathcal{D}}}) \big], \text{ s.t. } |\hat{\mathcal{D}}| \leq B, \quad (1)$$

where $\boldsymbol{x}$ follows the distribution $P$ of the testing dataset, and $\mathcal{L}_{\text{CE}}$ is the cross-entropy loss. The objective is to construct a coreset that enables the model to effectively discriminate among classes during testing. Accordingly, the class set $\mathcal{C}$ is identical between the training and testing datasets, in classification tasks.

**Coreset Selection for ReID** The goal of ReID is to retrieve images from the gallery set $\mathcal{G}$ that share the same identity

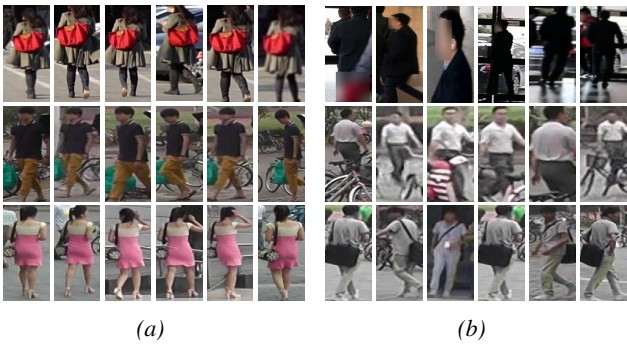

*(a)*          *(b)*

*Figure 5.* Sample images from classes with (a) low and (b) high diversity scores. Each row corresponds to a different identity.

as a given query image from the query set $\mathcal{Q}$, where each identity corresponds to a class. Note that the identity sets in the training dataset are disjoint from those in the query and gallery sets used for testing. Therefore, the strict requirement of consistent classes between training and testing, as in classification tasks, no longer applies in ReID. In the sequel, we reformulate coreset selection for ReID as a joint optimization problem consisting of: 1) finding an optimal class set $\hat{\mathcal{C}}^*$ from $\mathcal{C}$, and 2) constructing an optimal coreset $\hat{\mathcal{D}}^*(\hat{\mathcal{C}}^*)$ for a given $\hat{\mathcal{C}}^*$:

$$\hat{\mathcal{D}}^*(\hat{\mathcal{C}}^*) = \arg\min_{\hat{\mathcal{D}}(\hat{\mathcal{C}})} \mathbb{E}_{\substack{\boldsymbol{x}_i \sim P_{\mathcal{Q}} \\ \boldsymbol{x}_j \sim P_{\mathcal{G}}}} \left[ d(\boldsymbol{x}_i, \boldsymbol{x}_j; \phi_{\hat{\mathcal{D}}(\hat{\mathcal{C}})}) \mid y_i = y_j \right],$$

$$\text{s.t. } \hat{\mathcal{C}} \subseteq \mathcal{C}, \ \hat{\mathcal{D}}(\hat{\mathcal{C}}) \subset \mathcal{D}, \text{ and } |\hat{\mathcal{D}}(\hat{\mathcal{C}})| \leq B, \quad (2)$$

where $\boldsymbol{x}_i$ and $\boldsymbol{x}_j$ follow the distributions $P_{\mathcal{Q}}$ and $P_{\mathcal{G}}$ for the query and gallery sets, respectively, and $d(\cdot, \cdot)$ denotes a distance of features between $\boldsymbol{x}_i$ and $\boldsymbol{x}_j$ extracted by using $\phi_{\hat{\mathcal{D}}(\hat{\mathcal{C}})}$. This formulation aims to find both the optimal class set $\hat{\mathcal{C}}^*$ and the optimal coreset $\hat{\mathcal{D}}^*$ that together minimize the feature distance between query and gallery samples sharing the same identity. Figure 4 illustrates proposed two-stage CSOR framework composed of Diversity-driven Class Pruning (DCP) and Coverage-Prioritized Sampling (CPS).

### 3.2. Diversity-driven Class Pruning

For CSOR, it is permissible to select only a subset of classes, $\hat{\mathcal{C}} \subseteq \mathcal{C}$. As shown in Figure 2, selecting an optimal class set is crucial to improving intra-class diversity within $\hat{\mathcal{D}}$ under a limited storage budget $B$. To this end, we first remove classes with less diverse images, as they contribute least to learning robust and discriminative features. Specifically, we quantify an intra-class feature diversity score, $\delta^c$, for each class $c$ as the trace of the covariance matrix of the features of all samples in that class, which is given by

$$\delta^c = \text{Tr}(\text{Cov}(\mathcal{F}^c)), \quad (3)$$

where $\mathcal{F}^c$ denotes the set of features of all samples in class $c$ extracted by $\phi_D$. Since the proposed diversity score measures the total variation of features within a class, it effectively captures the intra-class diversity.

Figure 5 shows sample images in classes ranked from low to high diversity scores. We observe that samples in low-diversity classes (a) exhibit highly similar appearances, with only slight variations in background or pose. These samples are less informative and more redundant for learning discriminative features. In contrast, high-diversity classes (b) contain samples with significant variations in appearance, pose, lighting, and occlusion, making them more informative. However, the wide variation in features makes these classes harder to learn, and they may include outlier samples that are difficult to associate with a specific class.

All classes from $\mathcal{C}$ are sorted in ascending order of their diversity scores, and the classes with the lowest scores, corresponding to the bottom $\alpha$ proportion, are removed, resulting in a pruned class set $\hat{\mathcal{C}}$.

### 3.3. Coverage-Prioritized Sampling

We allocate a given storage budget $B$ to select images from the resulting class set $\hat{\mathcal{C}}$. Specifically, we adopt an easy-to-hard strategy inspired by curriculum learning (Bengio et al., 2009), which suggests that models learn more effectively when they are first exposed to easier examples before progressing to harder ones. Classes with low $\delta^c$ are considered easy ones since they exhibit more homogeneous features within the class, and those with high $\delta^c$ are hard ones due to their diverse features. Hence we process classes in $\hat{\mathcal{C}}$ in ascending order of their diversity scores to prioritize easy classes for budget allocation.

For each class $c \in \hat{\mathcal{C}}$, our objective is to select a subset $\hat{\mathcal{F}}^c \subseteq \mathcal{F}^c$ that is highly representative of $\mathcal{F}^c$. To achieve this, we utilize the facility location function (Owen & Daskin, 1998) to quantify the coverage of feature set, given by

$$\rho(\hat{\mathcal{F}}^c) = \frac{1}{|\mathcal{F}^c|} \sum_{\boldsymbol{f}_j \in \mathcal{F}^c} \max_{\boldsymbol{f}_i \in \hat{\mathcal{F}}^c} s_{ij}, \quad (4)$$

where $s_{ij} = \max\left(0, \cos(\boldsymbol{f}_i, \boldsymbol{f}_j)\right)$ and $\cos(\boldsymbol{f}_i, \boldsymbol{f}_j)$ measures the cosine similarity between two features of $\boldsymbol{f}_i$ and $\boldsymbol{f}_j$. In essence, $\rho$ measures how well the selected subset $\hat{\mathcal{F}}^c$ covers the entire set $\mathcal{F}^c$ by aggregating the similarity of each feature in $\mathcal{F}^c$ to its nearest neighbor within $\hat{\mathcal{F}}^c$. Finding the optimal subset $\hat{\mathcal{F}}^c$ that maximizes $\rho(\hat{\mathcal{F}}^c)$ in (4) is an NP-hard problem. However, we approximate the optimal solution by using a greedy algorithm, since $\rho$ is a monotone and submodular function (Nemhauser et al., 1978). In practice, for a given $\hat{\mathcal{F}}^c$, we compute the marginal gain in coverage, denoted as $\Delta(\boldsymbol{f}_k \mid \hat{\mathcal{F}}^c) = \rho(\hat{\mathcal{F}}^c \cup \{\boldsymbol{f}_k\}) - \rho(\hat{\mathcal{F}}^c)$, of a feature $\boldsymbol{f}_k \in \mathcal{F}^c \setminus \hat{\mathcal{F}}^c$, which is given by

$$\Delta(\boldsymbol{f}_k \mid \hat{\mathcal{F}}^c) = \frac{1}{|\mathcal{F}^c|} \sum_{\boldsymbol{f}_j \in \mathcal{F}^c} \max\left(0, s_{kj} - \max_{\boldsymbol{f}_i \in \hat{\mathcal{F}}^c} s_{ij}\right). \quad (5)$$

We iteratively select $\boldsymbol{x}_k$ that maximizes the marginal gain

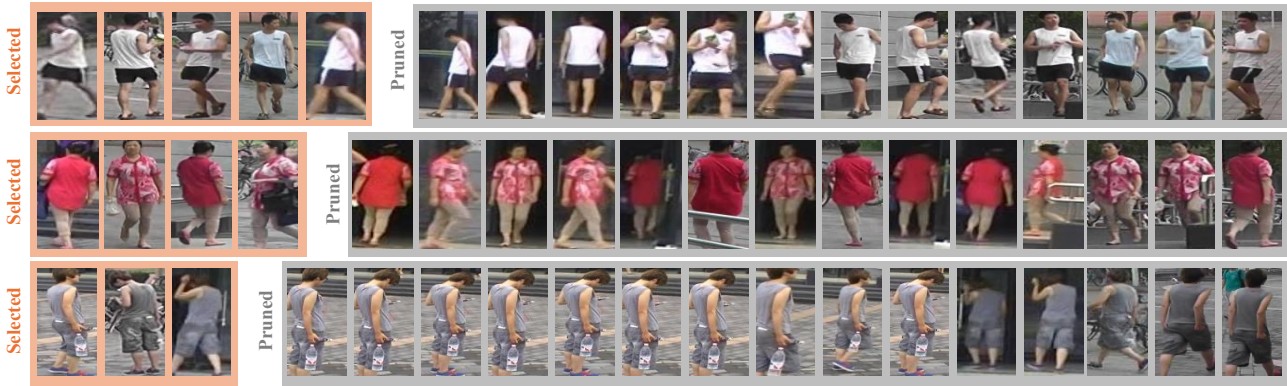

*Figure 6.* Results of CPS. The orange and gray colors indicate the selected and unselected samples, respectively. Each row indicates a different identity.

---

**Algorithm 1** Proposed CSOR Framework

**Require:** Original dataset $\mathcal{D}$, class set $\mathcal{C}$, feature extractor $\phi_{\mathcal{D}}$, class pruning ratio $\alpha$, storage budget $B$, sampling termination threshold $\tau$

**Ensure:** Coreset $\hat{\mathcal{D}}$, class subset $\hat{\mathcal{C}}$

1: $\hat{\mathcal{D}} \leftarrow \emptyset$
2: // Stage 1 Diversity-driven Class Pruning
3: **for** $c \in \mathcal{C}$ **do**
4: $\quad \mathcal{F}^c \leftarrow \{\phi_{\mathcal{D}}(\boldsymbol{x}_i) \mid y_i = c\}$
5: $\quad \delta^c \leftarrow \text{Tr}(\text{Cov}(\mathcal{F}^c))$ $\quad \triangleright$ Diversity Measurement
6: **end for**
7: $\hat{\mathcal{C}} \leftarrow \text{Top } (1-\alpha) \text{ fraction of } \mathcal{C} \text{ by } \delta^c$ $\quad \triangleright$ Class Pruning
8: Sort $\hat{\mathcal{C}}$ by $\delta^c$ in ascending order
9: // Stage 2: Coverage-Prioritized Sampling
10: **for** $c \in \hat{\mathcal{C}}$ **do**
11: $\quad \hat{\mathcal{F}}^c \leftarrow \emptyset$
12: $\quad$ **while** $|\hat{\mathcal{D}}| < B$ **do**
13: $\quad\quad \boldsymbol{f}^\star \leftarrow \arg\max_{\boldsymbol{f} \in \mathcal{F}^c \setminus \hat{\mathcal{F}}^c} \Delta(\boldsymbol{f} \mid \hat{\mathcal{F}}^c)$
14: $\quad\quad$ **if** $\Delta(\boldsymbol{f}^\star \mid \hat{\mathcal{F}}^c) < \tau$ **then**
15: $\quad\quad\quad$ **break** $\quad\quad\quad \triangleright$ Redundancy Avoidance
16: $\quad\quad$ **end if**
17: $\quad\quad \hat{\mathcal{F}}^c \leftarrow \hat{\mathcal{F}}^c \cup \{\boldsymbol{f}^\star\}$
18: $\quad\quad \hat{\mathcal{D}} \leftarrow \hat{\mathcal{D}} \cup \{\boldsymbol{x}^\star\}$
19: $\quad$ **end while**
20: $\quad$ **if** $|\hat{\mathcal{D}}| \geq B$ **then**
21: $\quad\quad$ **break** $\quad\quad\quad\quad\quad \triangleright$ Budget Exhausted
22: $\quad$ **end if**
23: **end for**
24: **return** $\hat{\mathcal{D}}, \hat{\mathcal{C}}$

---

$\Delta(\boldsymbol{f}_k \mid \hat{\mathcal{F}}^c)$, which is then included to update $\hat{\mathcal{D}}$.

While this greedy update scheme ensures that $\rho(\hat{\mathcal{F}}^c)$ incrementally covers the entire feature space of $\mathcal{F}^c$, samples selected later may contribute little to improving coverage due to redundancy. To mitigate such redundant sampling, we monitor the maximum marginal gain at each iteration

and terminate the sampling process for a class when this gain falls below a threshold $\tau$. We then proceed to the next class in priority order until the allocated storage budget is exhausted.

Figure 6 visualizes the results of CPS in terms of the selected and unselected samples of several classes. The selected ones are presented in the order of their selection by the greedy algorithm, which iteratively maximizes the marginal coverage gain in Eq. (5). We observe that CPS successfully selects diverse samples that cover various appearances within each class, such as different poses, resolutions, and backgrounds. Conversely, the unselected samples are deemed redundant because their marginal gains fall below the threshold $\tau$. These samples are often visually similar to ones already selected, demonstrating that our method effectively avoids redundancy and ensures an efficient use of the budget.

The overall algorithm is summarized in Algorithm 1.

## 4. Experimental Results

### 4.1. Experimental Setup

We evaluate the performance of the proposed method using three major person ReID datasets-Market1501 (Zheng et al., 2015), MSMT17-V2 (Wei et al., 2018), and CUHK03-NP (Zhong et al., 2017)-and one vehicle ReID dataset, VeRi-776 (Liu et al., 2016). The performance is measured by mean Average Precision (mAP) and Rank-1 accuracy (R-1), which are standard evaluation metrics for ReID. More implementation details are described in Appendix D.

### 4.2. Performance Comparison

**Quantitative ReID Performance** We compare the performance of the proposed method against several baselines, including six state-of-the-art CS methods: *Uniform*, *Random*, *Forgetting* (Toneva et al., 2019), *AUM* (Pleiss et al., 2020), *EL2N* (Paul et al., 2021), *GraNd* (Paul et al., 2021),

| Dataset | Market1501 | | | | | | MSMT17-V2 | | | | | | CUHK03-NP | | | | | |
|---|---|---|---|---|---|---|---|---|---|---|---|---|---|---|---|---|---|---|
| Budget (ratio) | 500 (3.8%) | | 1,000 (7.7%) | | 1,500 (11.6%) | | 500 (1.5%) | | 1,000 (3.1%) | | 1,500 (4.6%) | | 500 (6.8%) | | 1,000 (13.6%) | | 1,500 (20.4%) | |
| Metric | mAP | R-1 | mAP | R-1 | mAP | R-1 | mAP | R-1 | mAP | R-1 | mAP | R-1 | mAP | R-1 | mAP | R-1 | mAP | R-1 |
| Whole | 87.2 | 94.3 | 87.2 | 94.3 | 87.2 | 94.3 | 60.6 | 81.2 | 60.6 | 81.2 | 60.6 | 81.2 | 75.7 | 78.0 | 75.7 | 78.0 | 75.7 | 78.0 |
| Uniform | 14.7 | 31.7 | 42.3 | 63.8 | 65.3 | 82.7 | 2.0 | 6.2 | 3.1 | 9.2 | 16.6 | 21.4 | 2.7 | 2.0 | 20.9 | 19.3 | _46.0_ | _46.5_ |
| Random | 37.0 | 58.3 | 54.5 | 73.3 | 65.7 | 83.1 | 7.6 | 18.7 | 15.4 | 32.1 | 21.4 | 40.8 | 15.5 | 13.4 | 35.3 | _36.3_ | 45.2 | 45.9 |
| Forgetting | 30.5 | 52.2 | 49.1 | 71.6 | 60.5 | 80.8 | 2.8 | 9.4 | 4.8 | 13.9 | 6.3 | 18.0 | 12.0 | 10.8 | 29.7 | 28.7 | 42.3 | 42.8 |
| AUM | _49.2_ | _71.8_ | 57.4 | 78.0 | 62.6 | 81.0 | _13.2_ | _32.5_ | _17.4_ | _39.5_ | 19.2 | 41.2 | _24.6_ | _22.9_ | 34.4 | 34.6 | 43.7 | 45.1 |
| EL2N | 46.2 | 70.2 | _59.4_ | _80.4_ | 65.9 | _84.6_ | 4.5 | 13.0 | 9.8 | 9.8 | 11.5 | 28.0 | 16.8 | 14.6 | 31.1 | 31.2 | 42.4 | 43.2 |
| GraNd | 38.1 | 61.0 | 55.2 | 75.7 | 64.4 | 82.5 | 8.4 | 22.1 | 15.2 | 34.1 | 18.1 | 38.2 | 11.8 | 10.4 | 28.3 | 26.9 | 40.8 | 39.7 |
| Moderate | 36.3 | 57.8 | 55.8 | 75.1 | 65.8 | 82.2 | 8.3 | 20.2 | 15.5 | 33.6 | _22.2_ | _43.3_ | 16.4 | 15.1 | _35.9_ | 34.5 | 45.1 | 46.2 |
| CCS | 43.0 | 64.2 | 57.2 | 76.3 | 64.6 | 81.8 | 10.3 | 24.7 | 16.6 | 35.1 | 21.2 | 41.9 | 17.8 | 15.4 | 35.6 | 35.8 | 44.2 | 44.8 |
| Proposed | **56.2** | **76.5** | **67.1** | **83.8** | **71.7** | **86.7** | **15.9** | **34.6** | **26.2** | **50.2** | **29.3** | **52.0** | **30.3** | **30.6** | **43.1** | **44.6** | **49.5** | **51.1** |

*Table 1.* Comparison of person ReID performance evaluated on Market1501, MSMT17 and CUHK03-NP datasets. *Budget* denotes the number of selected images, and *ratio* represents the proportion of selected images to the total number of images in the original dataset. The best and second-best performances are boldfaced and underlined, respectively.

| Dataset | VeRi-776 | | | | | |
|---|---|---|---|---|---|---|
| Budget (Ratio) | 500 (1.3%) | | 1,000 (2.6%) | | 1,500 (4.0%) | |
| Metric | mAP | R-1 | mAP | R-1 | mAP | R-1 |
| Whole | 79.1 | 96.7 | 79.1 | 96.7 | 79.1 | 96.7 |
| Uniform | 13.0 | 33.1 | 32.2 | 57.4 | _43.2_ | 70.5 |
| Random | 24.9 | 52.1 | 34.8 | 64.2 | 41.9 | 70.7 |
| Forgetting | 10.9 | 30.0 | 23.0 | 52.8 | 29.2 | 61.9 |
| AUM | 23.2 | 56.8 | 29.9 | 65.6 | 33.4 | 68.5 |
| EL2N | 19.0 | 52.3 | 26.5 | 62.7 | 32.0 | 66.0 |
| GraNd | 22.2 | _57.3_ | 26.4 | 60.8 | 32.7 | 67.1 |
| Moderate | 21.4 | 49.6 | 33.8 | 62.5 | 41.5 | 69.5 |
| CCS | _26.7_ | 56.0 | _35.2_ | _67.3_ | 42.1 | **72.4** |
| Proposed | **33.7** | **60.7** | **41.9** | **68.0** | **45.8** | _71.6_ |

*Table 2.* Comparison of vehicle ReID performance evaluated on VeRi-776 dataset.

| Budget | 1,000 | | | 1,500 | | |
|---|---|---|---|---|---|---|
| Method | # Classes | SPC | MMD($\downarrow$) | # Classes | SPC | MMD($\downarrow$) |
| Uniform | 751 | 1.33 | 2.39 | 751 | 2.00 | 1.27 |
| Random | 514 | 1.95 | 1.87 | 590 | 2.54 | 1.49 |
| Forgetting | 451 | 2.22 | 2.03 | 549 | 2.73 | 1.69 |
| AUM | 144 | 6.94 | 1.10 | 191 | 7.85 | 1.00 |
| EL2N | 208 | 4.81 | 0.71 | 280 | 5.36 | 0.70 |
| GraNd | 411 | 2.43 | 1.78 | 505 | 2.97 | 1.55 |
| Moderate | 488 | 2.05 | 1.79 | 579 | 2.59 | 1.56 |
| CCS | 356 | 2.81 | 1.73 | 416 | 3.61 | 1.45 |
| Proposed | 167 | 5.99 | **0.28** | 247 | 6.07 | **0.29** |

*Table 3.* Statistics of the coresets with 1,000 and 1,500 images selected from the Market1501 dataset, which contains 751 classes. SPC: Samples Per Class. MMD: average Maximum Mean Discrepancy between $\hat{\mathcal{F}}^c$ and $\mathcal{F}^c$ across the selected classes in $\hat{\mathcal{C}}$.

*Moderate* (Xia et al., 2023), and *CCS* (Zheng et al., 2023), evaluated at three different storage budgets: 500, 1,000, and 1,500 images. Tables 1 and 2 show the results of the person and vehicle ReID tasks, respectively. *Whole* means the performance of using the original training dataset, which serves as an upper bound. The *Uniform* method allocates an equal storage budget to all classes and selects samples randomly within each class. The *Random* method randomly selects samples from the entire dataset, regardless of class labels.

When the storage budget is highly limited, the *Uniform* method usually performs the worst, since it does not consider different characteristics of classes for CS. However, as the budget increases, the *Uniform* method achieves comparable performance, even surpassing existing CS methods (e.g., attaining the second-best performance on the CUHK03-NP and VeRi-776 datasets in terms of mAP). These results indicate that while intra-class diversity is critical for CS at low storage budgets, inter-class diversity becomes increasingly important once sufficient intra-class diversity is achieved at higher budgets. *AUM* achieves the second-best performance in most cases when the storage budget is highly limited (e.g.,

500 images). This is because *AUM* tends to prune samples that are likely hard while retaining easier ones that provide more informative cues for learning discriminative features under such limited conditions, consistent with our easy-to-hard strategy for budget allocation priority. On the other hand, the other CS methods fail to achieve notable performance improvement over the *Random* method in most cases, since they are developed primarily for classification tasks, ignoring intra-class diversity, which is crucial for ReID. In contrast, the proposed method consistently outperforms all the existing methods. Notably, on the Market1501 dataset, the proposed method achieves 82% of the mAP score of *Whole* while using only about 12% of the data.

Table 3 compares the statistics of the coresets containing 1,000 and 1,500 images selected from the Market1501 dataset, which consists of 751 classes. We provide the number of selected classes ($|\hat{\mathcal{C}}|$), the average number of samples per class (SPC), and the average Maximum Mean Discrepancy (MMD) distance between $\hat{\mathcal{F}}^c$ and $\mathcal{F}^c$, averaged over all the selected classes in $\hat{\mathcal{C}}$. The MMD value quantifies the difference of distributions between the two sets, where a lower score indicates that $\hat{\mathcal{F}}^c$ more faithfully represents $\mathcal{F}^c$. We see that the MMD values are highly correlated with the

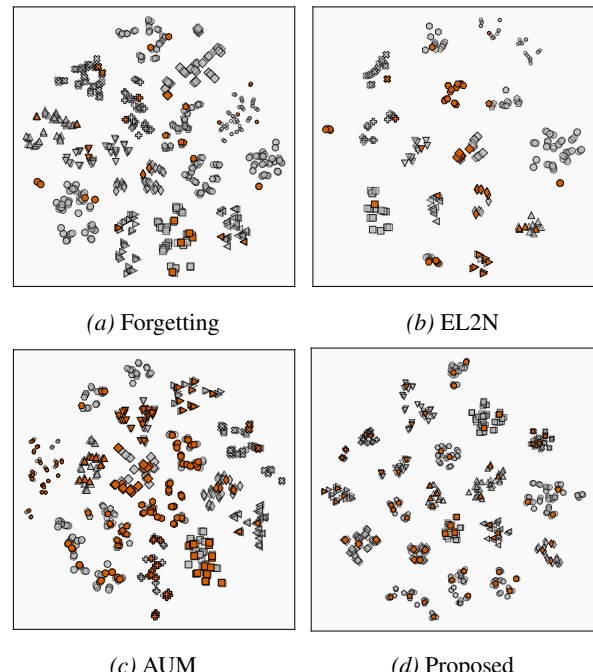

*(a)* Forgetting        *(b)* EL2N

*(c)* AUM        *(d)* Proposed

*Figure 7.* T-SNE (Maaten & Hinton, 2008) visualization of the 15 most frequently selected classes by each method on the Market1501 dataset with a budget of 1,000 images. Different marker shapes indicate different classes. The orange and gray colors represent selected and unselected samples, respectively.

| Budget | 3,000 | | 5,000 | | 7,000 | |
| Metric | mAP | R-1 | mAP | R-1 | mAP | R-1 |
| --- | --- | --- | --- | --- | --- | --- |
| Forgetting | 73.3 | 88.0 | 81.9 | 93.0 | **85.0** | **93.6** |
| EL2N | 75.8 | 90.2 | 80.9 | 92.1 | 83.2 | 93.1 |
| AUM | 70.6 | 86.1 | 77.0 | 88.9 | 80.8 | 91.6 |
| Proposed | **78.3** | **90.4** | **82.8** | 92.8 | 85.0 | 93.6 |

*Table 4.* Performance comparison under relaxed budgets (3,000, 5,000, and 7,000) on Market1501 dataset.

| Budget | 500 | | 1,000 | | 1,500 | |
| Metric | mAP | R-1 | mAP | R-1 | mAP | R-1 |
| --- | --- | --- | --- | --- | --- | --- |
| Random | 37.0 | 58.3 | 54.5 | 73.3 | 65.7 | 83.1 |
| w/ DCP | 41.4 | 62.3 | 58.0 | 77.9 | 67.8 | 84.9 |
| w/ CPS | 54.1 | **75.6** | 64.3 | 81.7 | 69.7 | 85.3 |
| Proposed | **54.8** | 75.2 | **67.1** | **83.8** | **71.7** | **86.7** |

*Table 5.* Performance of the proposed Diversity-driven Class Pruning (DCP) and Coverage-Prioritized Sampling (CPS), evaluated on the Market1501 dataset.

ReID performance, indicating that better coverage of intra-class diversity leads to improved ReID performance. Under 1,000 images budget, we observe that *GraNd*, *Moderate*, and *CCS* select more SPC than *Random* with marginally better MMD scores, correlating with their limited improvement in ReID performance. In contrast, the proposed method selects significantly fewer classes achieving a much higher SPC, and achieves the best ReID performance with the lowest MMD score. It is noteworthy that, although the proposed method yields fewer SPC than *AUM* (5.99 vs. 6.94), it achieves a substantially lower MMD score (0.28 vs. 1.10). This demonstrates that our proposed method allocates limited storage budget more effectively to capture sufficient intra-class diversity, compared with the existing methods.

To further verify the scalability of the proposed method, we compare the performance under relaxed budget constraints (3,000, 5,000, and 7,000) on Market1501 dataset, as shown in Table 4. As the budget increases, the performance gap relative to whole-dataset training diminishes. In particular, our method achieves 97.5% of the performance obtained with whole-dataset training in terms of mAP score by utilizing only 54.1% of the entire data. Moreover, our method quickly recovers full-dataset performance and consistently outperforms all baselines across varying budgets.

**Visualization of Feature Distribution** Figure 7 compares

the T-SNE (Maaten & Hinton, 2008) plots of feature distributions for the 15 most frequently selected classes by several methods on the Market1501 dataset with a budget of 1,000 images, where different marker shapes indicate different classes. We highlight the actually selected samples in orange, while unselected samples are shown in gray. We see that *Forgetting* often selects hard samples near the boundaries of each class, while ignoring relatively easy samples located near the class centers. *EL2N* also exhibits a similar tendency while allocating more budget for specific identities. *AUM* selects an excessive number of samples per class, possibly resulting in suboptimal allocation due to redundancy. In contrast, the proposed method effectively selects diverse samples from all classes, leading to a more balanced and representative feature distribution.

### 4.3. Ablation Study

**Effect of DCP and CPS** We conduct an ablation study to verify the effectiveness of DCP and CPS for the proposed method. Table 5 shows the performance evaluated on the Market1501 dataset at three different budgets. 'w/ DCP' prunes classes with low diversity scores and then selects samples randomly from the remaining classes. 'w/ CPS' assigns the storage budget for all classes without performing DCP, according to the easy-to-hard class priority (low to high $\delta^c$). We observe that both DCP and CPS contribute to the performance gain over the *Random* method. Note that 'w/ CPS' achieves performance similar to the proposed method at highly limited budget of 500 images. This is because 'w/ CPS' allocates the entire budget mainly to easy classes, thereby effectively learning the model (Bengio et al., 2009). The key benefit of DCP becomes significant when the storage budget is high enough to also learn from the more informative hard classes. Finally, applying both achieves

| Budget | 500 | | 1,000 | | 1,500 | |
| Metric | mAP | R-1 | mAP | R-1 | mAP | R-1 |
|---|---|---|---|---|---|---|
| Sample-wise | 14.8 | 31.9 | 50.8 | 71.1 | 67.1 | 83.9 |
| Random class | 52.4 | 74.7 | 64.7 | 83.4 | 70.9 | 86.5 |
| Hard-to-easy | 44.2 | 68.3 | 60.5 | 80.9 | 68.3 | 86.4 |
| Proposed | **54.8** | **75.2** | **67.1** | **83.8** | **71.7** | **86.7** |

*Table 6.* Performance comparison of storage budget allocation strategies, evaluated on the Market1501 dataset.

| Budget | | 1,000 | | 1,500 | |
| Coreset | ReID | mAP | R-1 | mAP | R-1 |
|---|---|---|---|---|---|
| | SBS | 14.5 | 29.5 | 24.9 | 44.8 |
| | TransReID | 50.8 | 70.8 | 63.3 | 82.1 |
| Forgetting | CLIP-ReID | 44.1 | 68.1 | 57.7 | 78.4 |
| | SOLIDER | 55.7 | 77.9 | 70.3 | 87.7 |
| | SBS | 30.4 | 53.8 | 40.8 | 64.3 |
| | TransReID | 61.2 | 81.5 | 67.4 | 85.3 |
| EL2N | CLIP-ReID | 59.4 | 79.6 | 65.6 | 84.2 |
| | SOLIDER | 66.0 | 85.4 | 72.4 | 88.3 |
| | SBS | 31.8 | 52.4 | 38.3 | 59.6 |
| | TransReID | 59.0 | 79.1 | 63.7 | 82.2 |
| AUM | CLIP-ReID | 57.2 | 77.0 | 62.8 | 80.1 |
| | SOLIDER | 60.1 | 79.0 | 68.1 | 85.1 |
| | SBS | **39.7** | **62.2** | **49.2** | **70.9** |
| | TransReID | **69.0** | **85.5** | **73.7** | **88.5** |
| Proposed | CLIP-ReID | **65.3** | **83.6** | **70.8** | **86.6** |
| | SOLIDER | **70.4** | **86.8** | **76.0** | **89.5** |

*Table 7.* Comparison of the generalization performance with four state-of-the-art ReID methods: SBS (Luo et al., 2019), TransReID (He et al., 2021), CLIP-ReID (Li et al., 2023), and SOLIDER (Chen et al., 2023). The performance is evaluated on Market1501 dataset with budgets of 1,000 and 1,500 images.

the best performance, demonstrating the effectiveness and complementary nature of each component.

**Effect of Budget Allocation Strategy** We additionally conduct comparative experiments of four strategies of storage budget allocation: 1) 'Sample-wise' allocates the budget starting from the samples with the highest marginal gains $\Delta$ in (5), across all classes without class priority; 2) 'Random class' first selects classes randomly and then selects samples within each class until the sampling termination condition is met; 3) Hard-to-easy, which prioritizes classes with high $\delta^c$; and 4) the proposed method that prioritizes classes with low $\delta^c$ (easy-to-hard). Table 6 shows that the proposed easy-to-hard approach consistently achieves the best performance, while 'Sample-wise' strategy performs the worst. 'Sample-wise' behaves similarly to the *Uniform* method, as it tends to allocate the budget evenly across all classes by selecting samples solely based on their marginal gains, ignoring class characteristics. Note that the performance gap between Hard-to-easy approach and the proposed method gradually diminishes as the available budget increases. This suggests

| Source → Target | Forgetting | AUM | EL2N | Proposed |
|---|---|---|---|---|
| MA → CU | 5.4 / 4.2 | 13.5 / 12.5 | 13.6 / 12.0 | **17.4 / 16.3** |
| MA → MS | 6.7 / 20.6 | 10.3 / 28.4 | 11.4 / 29.8 | **11.5 / 30.1** |
| MS → CU | 1.8 / 1.2 | 1.9 / 0.9 | 8.3 / 7.9 | **15.0 / 13.5** |
| MS → MA | 8.3 / 21.6 | 10.1 / 26.2 | 27.1 / 51.1 | **35.3 / 59.5** |

*Table 8.* Comparison of cross-dataset performance (mAP / R-1 %) under 1,000-image budget, where MA, CU, and MS indicate Market1501, CUHK03, and MSMT17-V2 datasets, respectively.

that, with a sufficiently high storage budget, learning from hard classes also becomes advantageous. The proposed easy-to-hard approach benefits from this as well, since it incrementally incorporates hard-class samples into the coreset after prioritizing and selecting the easy classes.

### 4.4. Generalization Performance

**Cross-architecture Generalization** Since our coreset selection relies on features from a specific extractor ($\phi_D$, ViT-B/16 in our case), it is crucial to verify that the coreset is not overfitted to this architecture and generalizes well to other ReID models with different backbones. To verify the generalization capability, we validate the performance when the coreset is used to train other ReID models with different backbones. We conduct experiments on the Market1501 dataset with storage budgets of 1,000 and 1,500 images, using the coresets generated by *Forgetting*, *EL2N*, *AUM*, and the proposed method to train several state-of-the-art ReID models. These models utilize diverse backbones: SBS (Luo et al., 2019) (ResNet (He et al., 2016)), TransReID (He et al., 2021) and CLIP-ReID (Li et al., 2023) (ViT (Dosovitskiy, 2021)), and SOLIDER (Chen et al., 2023) (Swin Transformer (Liu et al., 2021)). As shown in Table 7, the coreset generated by the proposed method consistently achieves the best performance across all ReID models, irrespective of their architecture. This suggests that our method captures generalizable data characteristics rather than overfitting to specific architectures, thereby ensuring robust generalization across diverse model backbones.

**Cross-dataset Generalization Performance** We also evaluate the cross-dataset generalization ability of the proposed method. As shown in Table 8, under a 1,000-image budget, our method consistently outperforms all compared baselines. We attribute this strong generalization to our selection strategy: rather than approximating global distributions, our method minimizes redundancy while preserving intra-class diversity, thereby encouraging domain-agnostic features and mitigating overfitting.

### 4.5. Hyperparameter Analysis

We analyze the performance variation with respect to hyperparameters. Figure 8 (a) shows the performance change according to the class pruning ratio $\alpha$. We observe that the proposed method achieves the best performance when $\alpha$ is

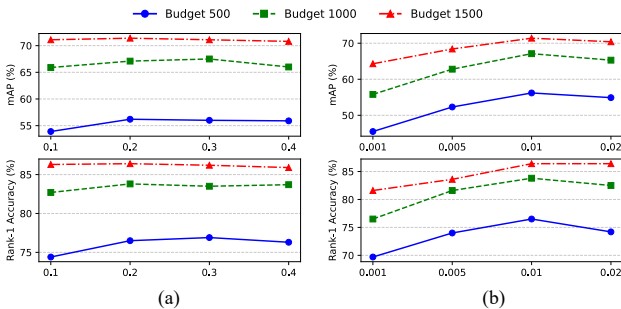

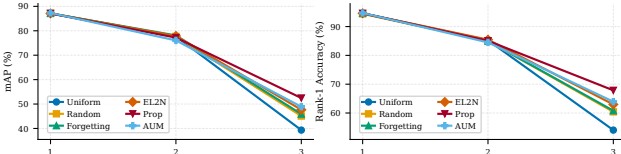

*Figure 8.* Performance variation according to (a) the class pruning ratio $\alpha$ and (b) the threshold $\tau$ for sampling termination. The performance is evaluted on the Market1501 dataset.

*Figure 9.* Performance comparison in the CL scenario in terms of mAP (left) and Rank-1 accuracy (right).

set to approximately 0.2. This implies that about 20% low diversity classes are indeed less informative and redundant for ReID training. Figure 8 (b) illustrates the performance change according to the threshold $\tau$ for sampling termination. The results indicate that a too-small $\tau$ leads to performance degradation, as it allows the excessive selection of redundant samples. Consequently, increasing $\tau$ improves performance up to a certain point, peaking at $\tau = 0.01$, which is the value selected for our experiments.

### 4.6. Applications

To demonstrate the practical utility of CSOR, we evaluate its effectiveness in continual learning (Wang et al., 2024; Shin et al., 2017) and neural architecture search (Elsken et al., 2019; Shim et al., 2021) settings for ReID.

**Continual Learning** In Continual Learning (CL), where models learn from sequentially arriving datasets, a coreset serves as a memory buffer under a limited budget, enabling rehearsal to mitigate catastrophic forgetting. We evaluate CL performance under a fixed per-stage memory budget. Specifically, we sequentially train a ViT-B/16 ReID model using cross-entropy and triplet losses, assuming that the Market-1501, CUHK03, and MSMT17 datasets arrive in order. After training on each dataset, we construct and store a 500-image coreset for that stage using each CS method, and reuse the stored coresets for rehearsal in subsequent stages. Figure 9 reports the average mAP and Rank-1 accuracy at each stage. Our method achieves the best final performance (52.5% mAP / 67.9% Rank-1), outperforming the second-best *AUM* (48.9% / 63.9% Rank-1). These results suggest that our coreset provides a stronger proxy for past data under strict storage constraints.

| Method | Forgetting | EL2N | AUM | Proposed |
|---|---|---|---|---|
| Correlation | 0.29 | 0.57 | 0.64 | **0.71** |
| mAP (%) | 78.9 | 79.4 | 79.1 | **80.2** |

*Table 9.* Performance comparison in the NAS scenario in terms of Kendall's $\tau$ (ranking correlation) and best mAP (%).

**Neural Architecture Search** Neural architecture search (NAS) is another practical scenario where CS is valuable, as it enables efficient evaluation of candidate architectures under tight computational budgets. We evaluate eight ViT-B/16 variants by varying the number of Transformer layers, the MLP hidden dimension, and the stride. Each candidate is trained on a coreset for 10 epochs to obtain a coreset-based validation ranking, while the reference ranking is obtained by training on the full dataset for 30 epochs. We report Kendall's $\tau$ between the two rankings, where a higher value indicates better preservation of the full-data ranking of candidate architectures. As shown in Table 9, the proposed method achieves the highest correlation ($\tau = 0.71$) and the best performance (80.2% mAP) among the compared methods, indicating more reliable coreset-based NAS.

## 5. Conclusion

We introduced a novel problem, **Coreset Selection for Object Re-identification (CSOR)**, under limited storage budgets and proposed a two-stage framework to address it. The framework first prunes classes and then selects representative samples from the remaining ones. Specifically, we quantify the diversity of each class and remove classes with low diversity scores, which are less informative and highly redundant. We then process the remaining classes in an easy-to-hard order and greedily select samples to maximize feature coverage using a submodular facility location function. Experimental results on three person ReID datasets and one vehicle ReID dataset demonstrate that the proposed method consistently achieves state-of-the-art ReID performance compared to existing CS methods.

**Limitation and Future Direction** A primary limitation of the current framework is its reliance on annotated data. While several studies (Sorscher et al., 2022) have explored unsupervised CS using clustering-based pseudo-labels (e.g., K-means clustering), adapting such approaches to ReID tasks remains highly non-trivial. Due to the fine-grained visual distinctions inherent in ReID datasets, standard clustering algorithms often produce noisy and unreliable pseudo-labels. Consequently, constructing a robust coreset under such noisy conditions poses a fundamentally different challenge. Nevertheless, extending CSOR to fully label-free scenarios would significantly broaden its practical utility. We leave the development of unsupervised CSOR as an important direction for future work.

## Impact Statement

Formulating the task of Coreset Selection for Object Re-identification (CSOR) establishes a foundational framework for addressing the severe data redundancy inherent in ReID datasets. By introducing this new problem setting, our work provides a principled framework that jointly optimizes class pruning and sample selection, achieving substantially improved training efficiency and recognition performance compared to conventional data pruning approaches. Moreover, the proposed framework has practical implications for resource-constrained deep learning scenarios, including continual learning, mobile AI, and federated learning. By reducing storage requirements while preserving informative intra-class diversity, our method enables more scalable, efficient, and sustainable model training under strict memory and computation budgets.

## Acknowledgements

This work was supported in part by the National Research Foundation of Korea (NRF) grant funded by the [Ministry of Science and ICT (MSIT)] under Grant RS-2024-00392536, in part by the Institute of Information and Communications Technology Planning and Evaluation (IITP) grants funded by the Korean Government (MSIT), including the Leading Generative Artificial Intelligence (AI) Human Resources Development Program under Grant IITP-2025-RS-2024-00360227, in part by the Artificial Intelligence Graduate School Program [Ulsan National Institute of Science and Technology (UNIST)] under Grant RS-2020-II201336, and in part by the AI Star Fellowship Program (UNIST) under Grant RS-2025-25442824.

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

# A. Submodular Function

The proposed Coverage-Prioritized Selection (CPS) is based on the facility location function that satisfies monotonicity and submodularity.

**Definition 1 (Monotonicity (Nemhauser et al., 1978))** A set function $\rho : 2^{\mathcal{F}^c} \to \mathbb{R}$ is monotone if for any $\mathcal{A}^c \subseteq \mathcal{F}^c$, we have $\rho(\mathcal{A}^c) \leq \rho(\mathcal{A}^c \cup \{\boldsymbol{f}_k\})$ for any $\boldsymbol{f}_k \in \mathcal{F}^c \setminus \mathcal{A}^c$.

**Definition 2 (Submodularity (Nemhauser et al., 1978))** A set function $\rho : 2^{\mathcal{F}^c} \to \mathbb{R}$ is submodular if for any $\mathcal{A}^c \subseteq \mathcal{B}^c \subseteq \mathcal{F}^c$ and any $\boldsymbol{f}_k^c \in \mathcal{F}^c \setminus \mathcal{B}^c$, the marginal gain satisfies:

$$\Delta(\boldsymbol{f}_k \mid \mathcal{A}^c) \geq \Delta(\boldsymbol{f}_k \mid \mathcal{B}^c).$$

These properties guarantee that the greedy algorithm used in CPS can approximate the optimal solution.

# B. Derivation of Coverage Gain

The coverage gain $\Delta(\boldsymbol{f}_k \mid \hat{\mathcal{F}}^c)$ measures the increase in coverage when adding a new feature vector $\boldsymbol{f}_k$ to the selected set $\hat{\mathcal{F}}^c$. It is defined as the difference between the coverage with and without $\boldsymbol{f}_k$:

$$
\begin{aligned}
\Delta(\boldsymbol{f}_k \mid \hat{\mathcal{F}}^c) &= \rho(\hat{\mathcal{F}}^c \cup \{\boldsymbol{f}_k\}) - \rho(\hat{\mathcal{F}}^c) \\
&= \frac{1}{|\hat{\mathcal{F}}^c|} \sum_{\boldsymbol{f}_j \in \hat{\mathcal{F}}^c} \max_{\boldsymbol{f}_i \in \hat{\mathcal{F}}^c \cup \{\boldsymbol{f}_k\}} s_{ij} - \frac{1}{|\hat{\mathcal{F}}^c|} \sum_{\boldsymbol{f}_j \in \hat{\mathcal{F}}^c} \max_{\boldsymbol{f}_i \in \hat{\mathcal{F}}^c} s_{ij} \\
&= \frac{1}{|\hat{\mathcal{F}}^c|} \sum_{\boldsymbol{f}_j \in \hat{\mathcal{F}}^c} \left[ \max\left( \max_{\boldsymbol{f}_i \in \hat{\mathcal{F}}^c} (s_{ij}, s_{kj}) \right) - \max_{\boldsymbol{f}_i \in \hat{\mathcal{F}}^c} s_{ij} \right] \\
&= \frac{1}{|\hat{\mathcal{F}}^c|} \sum_{\boldsymbol{f}_j \in \hat{\mathcal{F}}^c} \max\left( 0, \quad s_{kj} - \max_{\boldsymbol{f}_j \in \hat{\mathcal{F}}^c} s_{ij} \right).
\end{aligned}
$$

# C. Specifications of Datasets

The detailed specifications are summarized in Table 10.

| Dataset | Market1501 | MSMT17-V2 | CUHK03-NP | VeRi-776 |
|---|---|---|---|---|
| # Train IDs | 751 | 1,041 | 767 | 576 |
| # Train Images | 12,936 | 32,621 | 7,368 | 37,778 |
| # Test IDs | 750 | 3,060 | 700 | 200 |
| # Test Images | 19,281 | 93,820 | 6,728 | 13,257 |

*Table 10.* Specifications of the Market1501, MSMT17-V2, CUHK03-NP, and VeRi-776 datasets.

# D. Implementation Details

We adopt the ViT-B/16 (Dosovitskiy, 2021) as the feature extractor $\phi_D$, and use the combination of the cross-entropy loss and triplet loss to train the model. Note that we employ $\phi_D$, which has been trained for only a single epoch. We set

the class pruning ratio $\alpha = 0.2$ and the sampling termination threshold $\tau = 0.01$. Other training hyperparameters, such as learning rate and epochs, follow the configuration from (He et al., 2021). All experiments are implemented using PyTorch and run on a single RTX-3090 GPU.

# E. Comparison of Computational Cost

Table 11 compares the computational cost to construct coreset of Market1501 dataset under 7,000-image budget. As shown in the table, the proposed method is the fastest among the compared baselines. *Forgetting* and *AUM* require tracking forgetting events and classification confidence throughout the entire training process (120 epochs), which takes approximately 4,000 seconds and imposes a substantial computational burden, limiting their practicality. In contrast, *EL2N* and the proposed method utilize a model from a very early stage of training (i.e., after only one epoch), significantly reducing the overall computation time. Furthermore, although our method employs a greedy algorithm for Coverage-Prioritized Sampling (CPS), this procedure is performed independently within each class. As a result, the per-class computation takes only a fraction of a second, leading to highly efficient overall processing time.

| | Forgetting | AUM | EL2N | Proposed |
|---|---|---|---|---|
| Time (s) | 4,014.1 | 3,973.9 | 72.5 | 66.5 |

*Table 11.* Comparison of computational costs on the Market-1501 dataset under a 7,000-image budget.

# F. Statistics on Selected Coresets

We report more statistics on the coresets selected by the proposed method and the other state-of-the-art CS methods in Table 12. We can observe that the proposed method maintains a low class-wise MMD which indicates that each class distribution in the coreset is similar to that in the original dataset. We also observe that *EL2N* achieves lower MMD score than our method on VeRi-776 dataset under storage budget of 1,500 images, however, with a significantly lower performance (32.0% vs. 45.8% mAP) as shown in Table 2 in the main paper. We attribute this phenomenon to the extremely low number of selected classes (92 out of 576 classes) by *EL2N*, which leads to a scarcity of inter-class diversity.

# G. Additional Visualizations

### G.1. Diversity Score Distributions

Figure 10 illustrates the diversity score distribution across classes of each dataset. The red dashed line denotes the threshold determined by the proposed Diversity-driven Class Pruning with $\alpha$, where the classes with diversity scores

below the threshold are pruned. We can observe that the proposed method effectively prunes classes and captures the majority of diversity in each dataset.

### G.2. Selected Samples according to $\tau$

Figure 11 visualizes the selected samples via CPS when $\tau = 0.2, 0.1,$ and $0.005$. We can observe that as $\tau$ decreases, the more samples are selected while resulting in redundancy among the selected samples. Conversely, when $\tau$ increases, fewer samples are selected, leading to a loss of diversity. Therefore, it is crucial to choose an appropriate value of $\tau$ to balance diversity and redundancy in the selected coreset.

### G.3. Feature Distributions of Whole Dataset

Figure 12 illustrates the feature distributions of the selected coresets (under storage budget of 1,000 images) and the remaining samples with respect to the whole dataset on Market1501 dataset. Orange and gray colors indicate the selected coreset and the remaining samples, respectively. Most methods seem to better represent the overall distribution compared to the proposed method, while fail to achieve high performance as shown in Table 1 in the main paper. This suggests that the performance of coreset for ReID is less influenced by the global distribution.

### G.4. More Class-wise Feature Distributions

Figure 13 provides more visualizations of feature distributions for the most frequently selected 15 classes on Market1501 dataset under storage budget of 1,000 images. We can observe that the proposed method effectively captures the class-wise coverage without redundancy compared to the other methods. This demonstrates that our method utilizes the given storage budget more efficiently, resulting in superior performance.

### G.5. More Low- and High-Diversity Class Samples

Figure 14 presents additional samples from classes with low and high diversity scores for each dataset. These examples illustrate that the proposed diversity score effectively captures intra-class diversity: classes with low diversity scores tend to contain samples with similar appearances, whereas classes with high diversity scores exhibit substantial variations in viewpoints, backgrounds, and lighting conditions.

| Dataset | Market1501 | | | | | | | | | MSMT17-V2 | | | | | | | | |
|---|---|---|---|---|---|---|---|---|---|---|---|---|---|---|---|---|---|---|
| Budget | 500 | | | 1,000 | | | 1,500 | | | 500 | | | 1,000 | | | 1,500 | | |
| Method | # Cls | SPC | MMD | # Cls | SPC | MMD | # Cls | SPC | MMD | # Cls | SPC | MMD | # Cls | SPC | MMD | # Cls | SPC | MMD |
| Uniform | 500 | 1.00 | 2.80 | 751 | 1.33 | 2.39 | 751 | 2.00 | 1.27 | 500 | 1.00 | 2.89 | 1,000 | 1.00 | 2.87 | 1,041 | 1.44 | 2.21 |
| Random | 349 | 1.43 | 2.33 | 514 | 1.95 | 1.87 | 590 | 2.54 | 1.49 | 371 | 1.35 | 2.50 | 572 | 1.75 | 2.15 | 710 | 2.11 | 1.91 |
| Forgetting | 265 | 1.89 | 2.32 | 451 | 2.22 | 2.03 | 549 | 2.73 | 1.69 | 294 | 1.70 | 2.64 | 464 | 2.16 | 2.36 | 591 | 2.54 | 2.11 |
| AUM | 89 | 8.62 | 1.24 | 144 | 6.94 | 1.10 | 191 | 7.85 | 1.00 | 58 | 8.62 | 1.13 | 104 | 9.62 | 1.21 | 129 | 11.63 | 1.00 |
| EL2N | 119 | 4.2 | 0.74 | 208 | 4.81 | 0.71 | 280 | 5.36 | 0.70 | 169 | 2.96 | 2.07 | 264 | 3.79 | 1.98 | 327 | 4.59 | 1.76 |
| GraNd | 272 | 1.84 | 2.12 | 411 | 2.43 | 1.78 | 505 | 2.97 | 1.55 | 261 | 1.92 | 2.29 | 428 | 2.34 | 2.10 | 544 | 2.76 | 1.90 |
| Moderate | 332 | 1.51 | 2.28 | 488 | 2.05 | 1.79 | 579 | 2.59 | 1.56 | 347 | 1.45 | 2.39 | 535 | 1.87 | 2.07 | 672 | 2.23 | 1.86 |
| CCS | 241 | 2.07 | 1.97 | 356 | 2.81 | 1.73 | 416 | 3.61 | 1.45 | 247 | 2.02 | 2.12 | 373 | 2.68 | 1.92 | 481 | 3.12 | 1.81 |
| Proposed | 90 | 5.56 | **0.29** | 167 | 5.99 | **0.28** | 247 | 6.07 | **0.29** | 108 | 4.63 | **0.50** | 207 | 4.83 | **0.46** | 301 | 4.98 | **0.45** |

| Dataset | CUHK03-NP | | | | | | | | | VeRi-776 | | | | | | | | |
|---|---|---|---|---|---|---|---|---|---|---|---|---|---|---|---|---|---|---|
| Budget | 500 | | | 1,000 | | | 1,500 | | | 500 | | | 1,000 | | | 1,500 | | |
| Method | # Cls | SPC | MMD | # Cls | SPC | MMD | # Cls | SPC | MMD | # Cls | SPC | MMD | # Cls | SPC | MMD | # Cls | SPC | MMD |
| Uniform | 500 | 1.00 | 2.72 | 767 | 1.30 | 2.25 | 767 | 2.00 | 1.28 | 500 | 1.00 | 2.92 | 576 | 1.74 | 1.82 | 576 | 2.60 | 1.14 |
| Random | 372 | 1.34 | 2.23 | 570 | 1.75 | 1.83 | 677 | 2.22 | 1.44 | 309 | 1.62 | 2.23 | 435 | 2.30 | 1.78 | 496 | 3.02 | 1.46 |
| Forgetting | 312 | 1.60 | 2.18 | 493 | 2.03 | 1.79 | 599 | 2.50 | 1.46 | 137 | 3.65 | 1.94 | 222 | 4.50 | 1.94 | 283 | 5.30 | 1.81 |
| AUM | 201 | 2.49 | 1.55 | 350 | 2.86 | 1.42 | 441 | 3.40 | 1.17 | 66 | 7.58 | 1.52 | 97 | 10.31 | 1.26 | 123 | 12.20 | 1.13 |
| EL2N | 212 | 2.36 | 1.82 | 328 | 3.05 | 1.52 | 417 | 3.60 | 1.24 | 48 | 10.42 | 0.71 | 77 | 12.99 | 0.72 | 92 | 16.30 | **0.52** |
| GraNd | 349 | 1.43 | 2.20 | 538 | 1.86 | 1.85 | 638 | 2.35 | 1.49 | 93 | 5.38 | 1.50 | 133 | 7.52 | 1.24 | 157 | 9.55 | 1.00 |
| Moderate | 360 | 1.39 | 2.22 | 526 | 1.90 | 1.73 | 636 | 2.36 | 1.45 | 321 | 1.56 | 2.31 | 424 | 2.36 | 1.78 | 480 | 3.13 | 1.43 |
| CCS | 324 | 1.54 | 2.08 | 497 | 2.01 | 1.71 | 588 | 2.55 | 1.37 | 201 | 2.49 | 2.02 | 294 | 3.40 | 1.75 | 350 | 4.29 | 1.61 |
| Proposed | 78 | 6.41 | **0.17** | 146 | 6.85 | **0.15** | 213 | 7.04 | **0.13** | 152 | 3.29 | **0.70** | 287 | 3.48 | **0.65** | 413 | 3.63 | 0.62 |

*Table 12.* Statistics of the coresets on the Market1501, MSMT17, CUHK03-NP, and VeRi-776 datasets under various storage budgets. # Cls and SPC denote the number of classes and the average number of samples per class, respectively. MMD denotes the class-wise maximum mean discrepancy between the original feature set and the selected coreset.

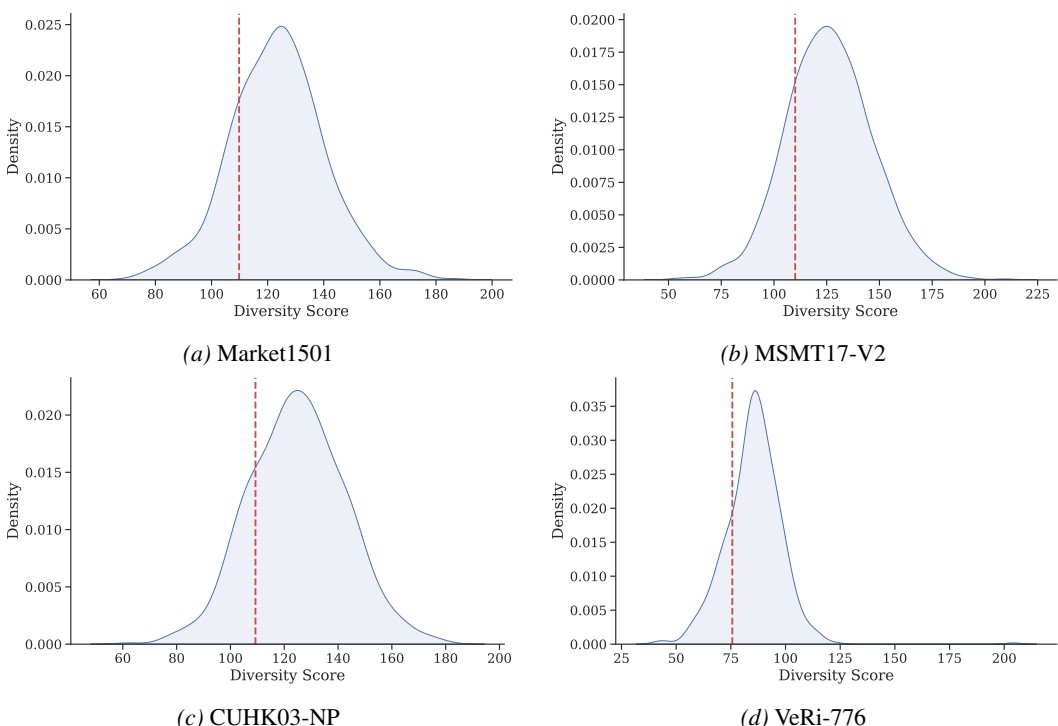

*(a)* Market1501

*(b)* MSMT17-V2

*(c)* CUHK03-NP

*(d)* VeRi-776

*Figure 10.* Diversity score distribution of each dataset. The red dashed line indicates the threshold that is pruned by the proposed Diversity-driven Class Pruning method with $\alpha$.

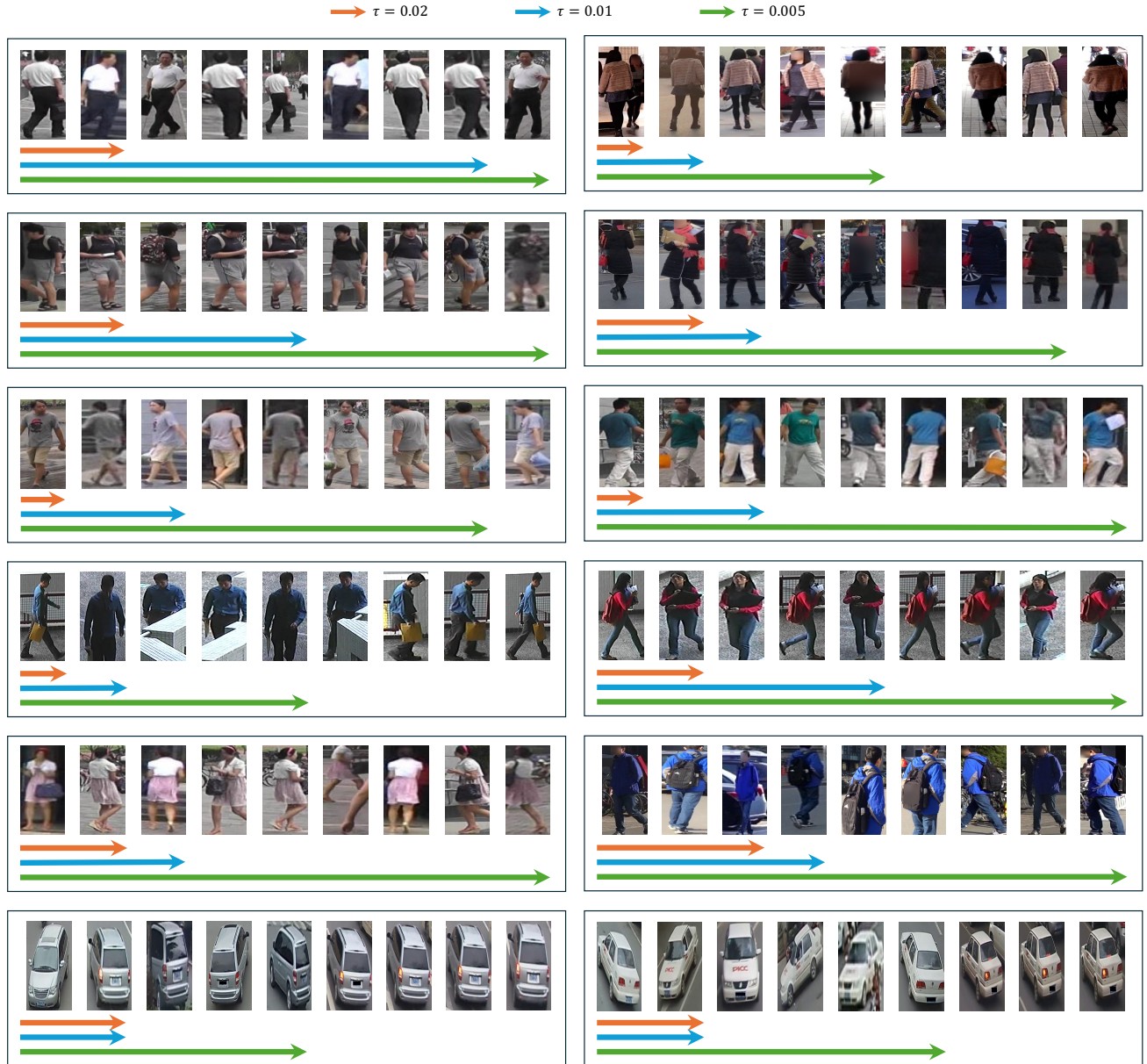

*Figure 11.* Visualization of selected samples via Coverage-Prioritized Sampling method according to different values of $\tau$. Each arrow indicates the selected samples for each $\tau$ value.

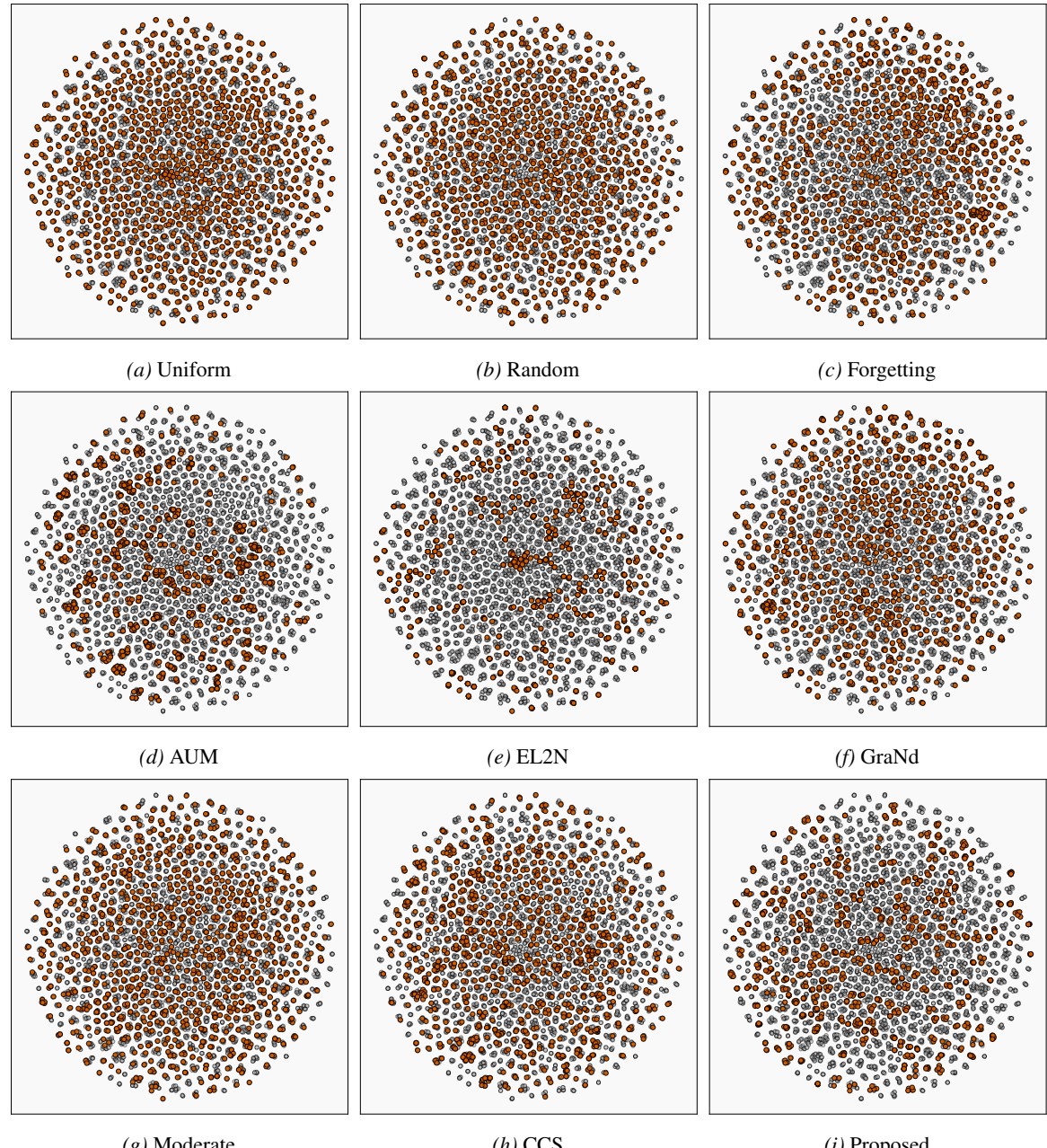

*(a)* Uniform

*(b)* Random

*(c)* Forgetting

*(d)* AUM

*(e)* EL2N

*(f)* GraNd

*(g)* Moderate

*(h)* CCS

*(i)* Proposed

*Figure 12.* Visualization of feature distribution with respect to the whole dataset on Market1501 dataset. Orange and gray colors indicate the selected coreset and the remaining samples, respectively.

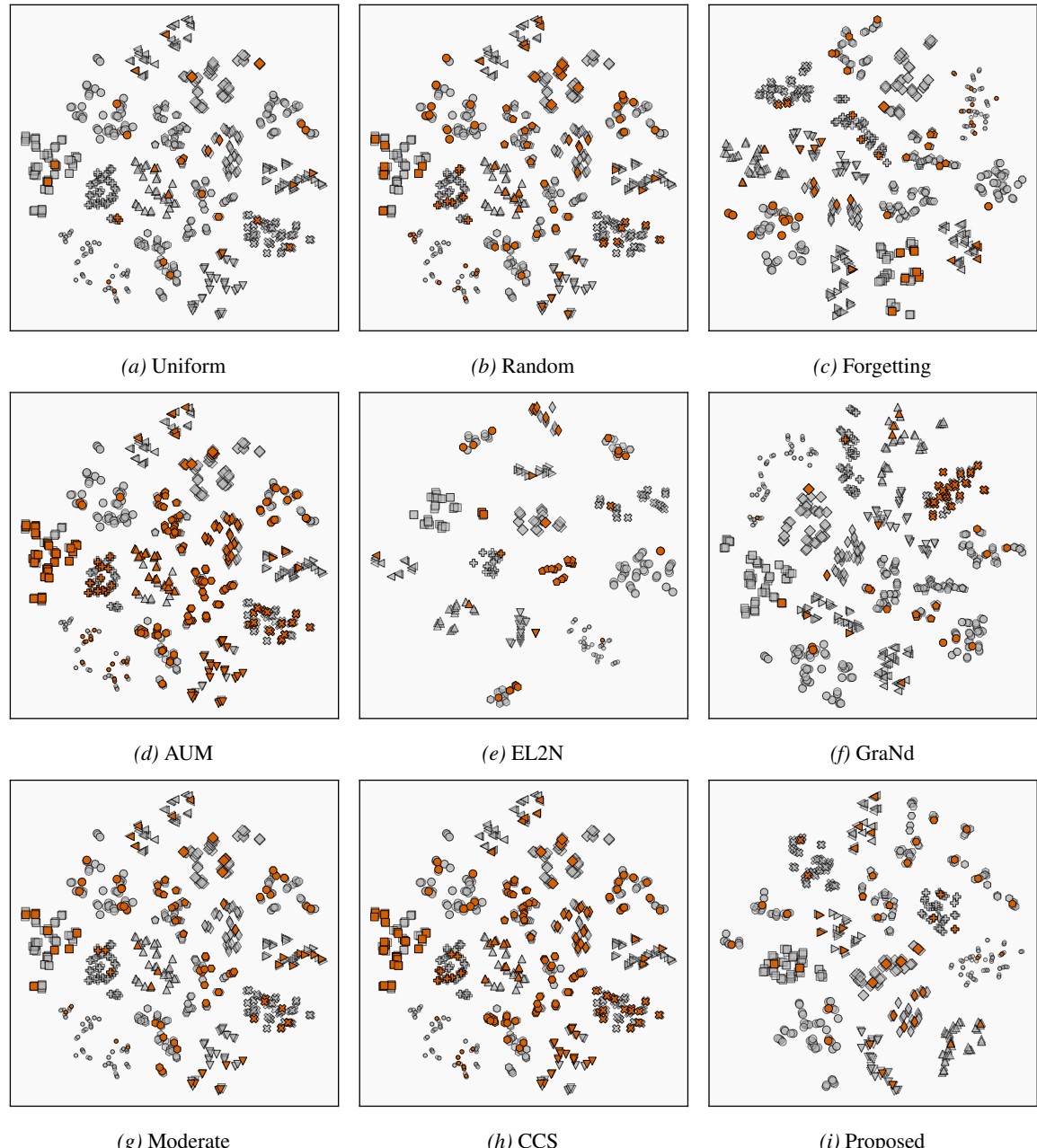

*(a)* Uniform      *(b)* Random      *(c)* Forgetting

*(d)* AUM      *(e)* EL2N      *(f)* GraNd

*(g)* Moderate      *(h)* CCS      *(i)* Proposed

*Figure 13.* Feature distribution visualization of the most frequently selected 15 classes on Market1501 dataset. Orange and gray colors indicate the selected coreset and the remaining samples, respectively.

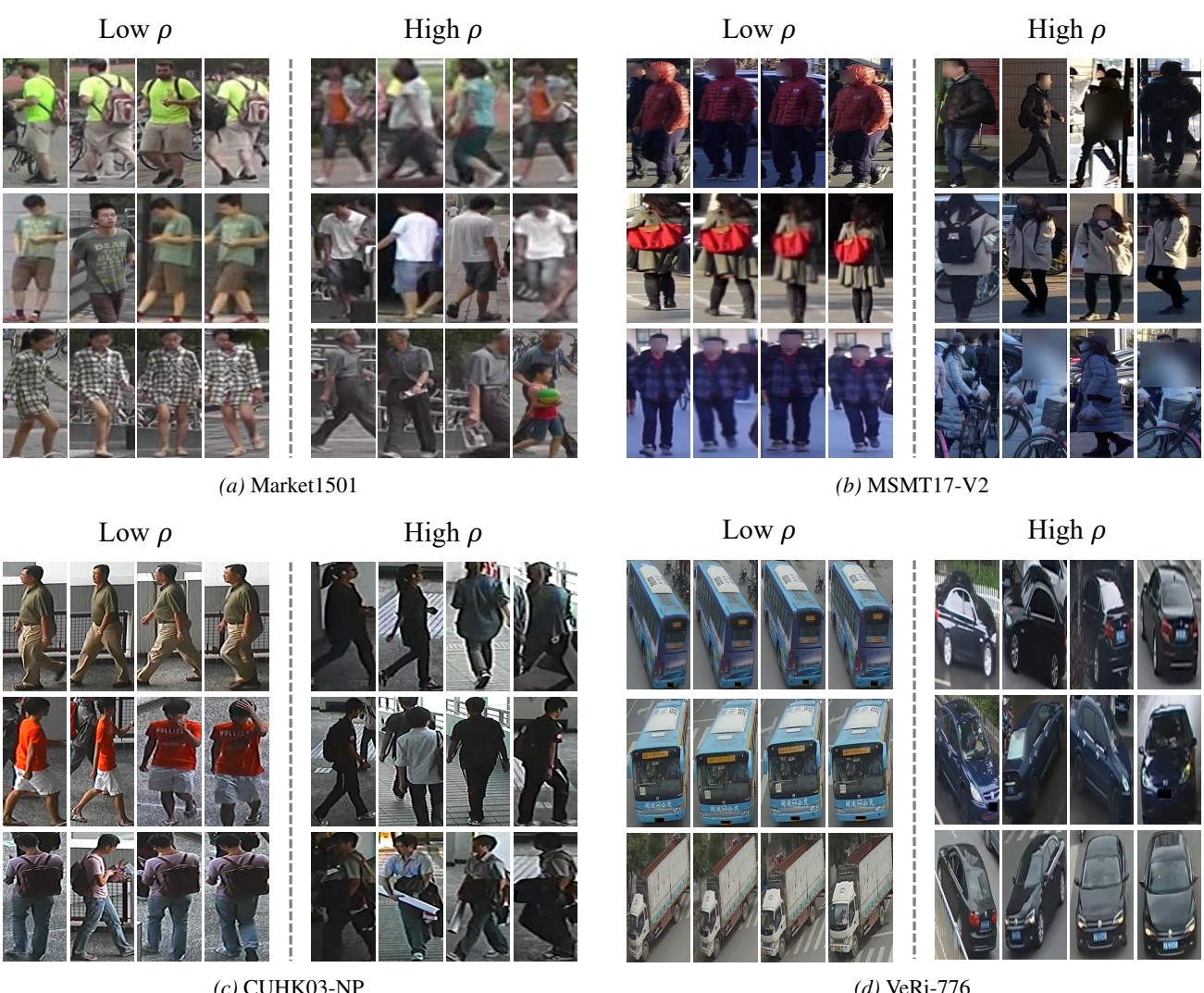

*(a)* Market1501

*(b)* MSMT17-V2

*(c)* CUHK03-NP

*(d)* VeRi-776

*Figure 14.* Low- and high-diversity score class samples of each dataset.

