# OpenReview forum: "CSOR: Coreset Selection for Object Re-identification via Class Pruning"
_ICML.cc/2026/Conference — ICML 2026 regular_

### Official Review · Reviewer_SzSW · 2026-03-09

**Soundness:** 4
**Presentation:** 4
**Significance:** 3
**Originality:** 3
**Overall Recommendation:** 5
**Confidence:** 4

**Summary:**

The authors introduce a two-stage Coreset Selection framework to extract a smaller yet representative subset of the training data to reduce the complexity of model training. They demonstrate that their proposed method outperforms exsting CS methods.

**Compliance With Llm Reviewing Policy:**

Affirmed.

**Final Justification:**

The paper presents a clear and well-motivated contribution. The writing is clear and the work is logically structured, and the contribution appears both original and meaningful. I therefore support acceptance.

**Key Questions For Authors:**

1) How does this compare in terms of computational expense to other methods?
2) Why focus only on object re-identification when presenting a more general framework?
3) Corset selection seems very similar to measuring data influence conceptually, as data influence can be used to retain the most important training samples. Have you looked into this?

Some relevant citations for question 3:

Koh, Pang Wei, and Percy Liang. Understanding Black-box Predictions via Influence Functions. In Proceedings of the 34th International Conference on Machine Learning (ICML), Proceedings of Machine Learning Research, vol. 70, pp. 1885–1894, 2017.

San Joaquin, Ayrton, Bin Wang, Zhengyuan Liu, Nicholas Asher, Brian Lim, Philippe Muller, and Nancy F. Chen. In2Core: Leveraging Influence Functions for Coreset Selection in Instruction Finetuning of Large Language Models. Findings of the Association for Computational Linguistics: EMNLP 2024, pp. 10324–10335, 2024.

**Limitations:**

Yes

**Strengths And Weaknesses:**

Strengths:

* Demonstrated improvements over existing, state of the art CS methods
* Employed several state-of-the-art time series models in the empirical evaluation.
* The methodology is rigorous and well justified.
* The paper is clearly written and logically structured.
* The authors effectively articulate the significance and motivation of the work.
* The contribution is original and novel.

Weaknesses

* While performance improvements are good, how does this compare in terms of computational expense to other methods?
* Coreset selection is not limited to object re-identification. Why focus only on this application when presenting a more general framework?

---

> ### Author Rebuttal · Authors · 2026-03-29
>
> **A1. Computational cost comparison**
>
> We measure the actual computation time (in seconds) required to select a coreset of size 7,000 images on the Market1501 dataset. All experiments are conducted under an identical hardware setup using a single RTX 3090 GPU.
>
> | | Proposed | EL2N | Forgetting | AUM |
> | :--- | :---: | :---: | :---: | :---: |
> | **Time (s)** | **66.54** | 72.51 | 4,014.09 | 3,973.87 |
>
> As shown in the table, the proposed method is the fastest among the compared baselines. Forgetting and AUM require tracking forgetting events and classification confidence throughout the entire training process (120 epochs), which takes approximately 4,000 seconds and imposes a substantial computational burden, limiting their practicality.
>
> In contrast, EL2N and the proposed method utilize a model from a very early stage of training (i.e., after only one epoch), significantly reducing the overall computation time. Furthermore, although our method employs a greedy algorithm for Coverage-Prioritized Sampling (CPS), this procedure is performed independently within each class. As a result, the per-class computation takes only a fraction of a second, leading to highly efficient overall processing time.
>
> We will include this computational cost analysis in the revised manuscript.
>
> **A2. Rationale of coreset selection for object ReID**
>
> As explained in the Introduction and illustrated in Figure 1 of the main paper, the fundamental distinction between datasets used for classification and ReID tasks lies in the relationship between the label spaces of the training and test sets.
>
> Standard image classification is a closed-set problem, where the training and test sets share the exact same label space. Consequently, conventional coreset selection methods for classification must strictly preserve samples from all existing classes to prevent catastrophic accuracy degradation for any omitted class.
>
> In contrast, ReID is an open-set retrieval task in which the training and test sets have completely disjoint label spaces. The objective is to learn a generalized feature representation rather than specific class boundaries. CSOR leverages this relaxed constraint by directly identifying and pruning less informative classes (via Diversity-driven Class Pruning). Therefore, CSOR fundamentally differs from general coreset selection methods for classification, as it expands the optimization scope from merely selecting "optimal samples" to jointly discovering "optimal classes."
>
> **A3. Differences from data influence**
>
> Thanks for this insightful question. While Data Influence and Coreset Selection (CS) share conceptual similarities—and certain CS baselines (e.g., *Forgetting*) implicitly leverage influence-like metrics by tracking top-influence data points—their core objectives and outcomes differ significantly.
>
> Data Influence primarily aims to measure the impact of an individual training sample on the model's predictions, which often requires high computational complexity with respect to all training samples. Furthermore, samples with high influence scores are typically instances with high loss values or extreme outliers. If a selection method purely targets these high-influence samples, the resulting subset tends to be visually biased (e.g., heavily occluded or mislabeled images). This leads to severe redundancy and low intra-class diversity, ultimately resulting in poor generalization performance (as observed with the *Forgetting* baseline).
>
> In contrast, Coreset Selection methods explicitly consider overall training efficiency and covering the global dataset distribution. Rather than greedily ranking individual sample importance, CS methods aim to find an optimal training subset that ensures sufficient *diversity* and *coverage* to accurately represent the entire dataset.

---

> > ### Author Rebuttal · Reviewer_SzSW · 2026-04-01
> >
> > Thank you for your thorough responses. I have no further concerns, and my original score of 5 (accept) remains unchanged.

---

> > > ### Author Response · Authors · 2026-04-02
> > >
> > > Thank you for your acknowledgment. We sincerely appreciate your valuable comments.

---

### Official Review · Reviewer_DbBG · 2026-03-13

**Soundness:** 3
**Presentation:** 3
**Significance:** 2
**Originality:** 3
**Overall Recommendation:** 4
**Confidence:** 3

**Summary:**

This paper introduces the novel problem of Coreset Selection for Object Re-identification (CSOR), addressing the unique characteristics of ReID datasets where training and testing label spaces are disjoint. The authors propose a two-stage framework that first uses Diversity-driven Class Pruning (DCP) to remove less informative classes, and then applies Coverage-Prioritized Sampling (CPS) to select diverse samples using a submodular facility location function. Experiments on multiple person and vehicle ReID datasets demonstrate that this approach outperforms existing coreset selection methods designed for standard classification.

**Compliance With Llm Reviewing Policy:**

Affirmed.

**Final Justification:**

The rebuttal addresses most of my concerns, especially the practical justification of the proposed setting.

**Key Questions For Authors:**

Please see the weaknesses discussed above.

**Strengths And Weaknesses:**

Strength:
1. The authors provide a compelling rationale for why standard classification coreset methods are suboptimal for ReID, effectively highlighting the critical need for intra-class diversity and the flexibility provided by disjoint label spaces.
2. The proposed two-stage framework (DCP and CPS) is technically sound and well-structured.

Weakness:
1. My primary concern lies in the practical justification of the proposed problem setting. The authors fail to adequately motivate the necessity of introducing Coreset Selection (CS) specifically for ReID tasks, nor do they provide sufficient evidence that storage budgets pose a critical bottleneck in standard ReID training pipelines.

(1) While the authors attempt to justify the setting by providing two application scenarios—Continual Learning and Neural Architecture Search (NAS) —these are entirely relegated to the Appendix (Appendix A). This leaves the main text lacking a convincing real-world anchor.

(2) Even if storage constraints are assumed to be an issue, the scale of the datasets used undermines the premise. For instance, the Market1501 dataset contains only around 32k training images, which is computationally trivial to process in its entirety on modern hardware. Demonstrating CS on such a small scale makes the problem setting feel artificial.

(3) The trade-off between storage efficiency and accuracy is too severe for practical adoption in ReID, a task that demands highly discriminative features. For example, under the proposed setting on Market1501, the mAP drops significantly to 71.7%.

2. The proposed method's reliance on labeled data limits its relevance in the current landscape of large-scale representation learning. In Table 6, the authors evaluate the generalization of their coreset on pre-trained foundation models like CLIP-REID and TransReID. However, in modern paradigms, data pruning and subset selection are predominantly valuable during the massive-scale unsupervised or self-supervised pre-training phase. During downstream fine-tuning, practitioners typically utilize the entire labeled dataset to push the limits of performance. The CSOR framework, particularly the Diversity-driven Class Pruning (DCP) stage, heavily relies on explicit identity (class) labels to compute intra-class diversity. It is unclear if or how this framework can be extended to an unsupervised ReID setting.

---

> ### Author Rebuttal · Authors · 2026-03-29
>
> **A1. Justification of the proposed problem setting**
>
> **Application scenarios**
>
> Thanks for a helpful comment. As detailed in Appendix A, CSOR can be used for practical resource-constrained applications such as Continual Learning and Neural Architecture Search, which can be used for edge devices (i.e., Jetson Nano [R1] that has only 16GB storage). As demonstrated in the literature [R2–R4], Continual ReID is one of the most practical applications, where systems must adapt to new domains while strictly limiting memory buffers to prevent catastrophic forgetting. In this scenario, CSOR efficiently constructs an optimal exemplar set for continual learning under a strict budget and outperforms baselines in preserving past knowledge, as shown in Fig. 9. As the reviewer suggested, we will integrate practical applications into the main text, and clarify the motivation of our research.
>
> [R1] https://developer.nvidia.com/embedded/jetson-nano
> [R2] "Lifelong person re-identification by pseudo task knowledge preservation." AAAI. 2022.
> [R3] "LSTKC: Long short-term knowledge consolidation for lifelong person re-identification." AAAI. 2024.
> [R4] "Diverse representations embedding for lifelong person re-Identification." IEEE TNNLS. 2025.
>
> **Motivation**
>
> In practical real-world applications, ReID data are usually obtained from multi-camera video streams and consist of a large number of image frames with subtle differences, which results in high temporal redundancy. Therefore, this leads to increased storage and training costs when storing and processing such data. Current representative benchmark datasets are curated versions of such massive raw data; for example, the Market1501 dataset is a curated version of the MARS dataset composed of over 1M images [R5]. We can use the proposed CSOR as an essential preprocessing tool to distill a highly informative, non-redundant subset from massive raw data at various scales depending on the application.
>
> [R5] "MARS: A video benchmark for large-scale person re-identification." ECCV. 2016.
>
> **Trade-off between storage efficiency and accuracy**
>
> We emphasize that the proposed method achieves a 71.7 mAP score while using only 11.6% of the full dataset—successfully reaching 82.2% of the performance obtained with the entire dataset—demonstrating exceptionally high data efficiency. Notably, this represents an 8.8% absolute improvement over the second-best baseline, *EL2N*.
>
> To further verify the scalability of the proposed method, we compare the performance (mAP / Rank-1 %) under increased budgets:
>
> | Budget | Forgetting | EL2N | AUM | Proposed |
> |:---:|:---:|:---:|:---:|:---:|
> | **3000** | 73.3 / 88.0 | 75.8 / 90.2 | 70.6 / 86.1 | **78.3 / 90.4** |
> | **5000** | 81.9 / 93.0 | 80.9 / 92.1 | 77.0 / 88.9 | **82.8 / 92.8** |
> | **7000** | 85.0 / 93.6 | 83.2 / 93.1 | 80.8 / 91.6 | **85.0 / 93.6** |
>
> As the budget increases, the performance gap relative to full-dataset training diminishes. In particular, our method achieves 97.5% of the performance obtained with full-dataset training, by utilizing only 54.1% of the entire data. Moreover, our method quickly recovers full-dataset performance and consistently outperforms all baselines. We will add this discussion in the revised version of the paper.
>
> **A2. Extension to unsupervised learning**
>
> Most existing coreset selection (CS) methods—including the baselines evaluated in this study—primarily operate in supervised settings and rely heavily on ground-truth class labels. Accordingly, our work also focuses on the supervised setting.
> While a few recent studies have explored unsupervised CS [R6] using pseudo-labels obtained via clustering (e.g., K-means), applying such approaches to ReID is non-trivial due to its fine-grained nature and inherent dataset characteristics, as illustrated in Figure 1. In practice, clustering-based pseudo-labels are often noisy and unreliable for ReID tasks. Consequently, constructing a robust coreset from such unreliable pseudo-labels constitutes a fundamentally different challenge that lies beyond the scope of this paper.
> Nevertheless, we acknowledge that extending CSOR to label-free scenarios would significantly enhance its practicality, and we consider this an important direction for future work, which will be further discussed in the revised manuscript.
>
> [R6] Sorscher, Ben, et al. "Beyond neural scaling laws: beating power law scaling via data pruning." NeurIPS, 2022.

---

> > ### Author Rebuttal · Reviewer_DbBG · 2026-04-05
> >
> > I thank the authors for addressing most of my concerns. I decided to adjust my score accordingly.

---

> > > ### Author Response · Authors · 2026-04-06
> > >
> > > We are pleased that your concerns have been addressed. We sincerely appreciate your constructive feedback and your consideration in raising the score.

---

### Official Review · Reviewer_vf2c · 2026-03-24

**Soundness:** 3
**Presentation:** 3
**Significance:** 2
**Originality:** 3
**Overall Recommendation:** 4
**Confidence:** 4

**Summary:**

This paper formulates Coreset Selection(CS) for Object Re-identification (CSOR) as a joint optimization problem to find both the optimal coreset and the optimal class subset. Based on the insight that intra-class diversity is a key factor for effective coreset construction for ReID, this paper proposes a two-stage framework, consisting of Diversity-driven Class Pruning (DCP)and Coverage-Prioritized Sampling (CPS) designed for ReID datasets. Experiments on three person ReID datasets and one vehicle ReID dataset show that the proposed method outperforms existing CS models.

**Compliance With Llm Reviewing Policy:**

Affirmed.

**Final Justification:**

My concerns have been addressed, so I adjust my score accordingly.

**Key Questions For Authors:**

1.	The motivation of applying coreset selection to ReID tasks is not fully convincing. Could the authors clarify in which practical scenarios coreset selection becomes necessary for ReID?

2.	The effectiveness of the proposed method appears to be dataset-dependent, with much larger performance drops on more challenging datasets. Could the authors discuss whether the selection strategy can be adapted or made more robust to different dataset properties?
No.
The paper does not sufficiently discuss the limitations of the proposed approach. In particular, the potential performance degradation when using a reduced training set, as well as the possible impact on generalization under domain shifts, are not adequately analyzed.

3.	Have the authors evaluated the method in a cross-dataset setting, e.g., training on one dataset and testing on another?

**Limitations:**

No.
The paper does not sufficiently discuss the limitations of the proposed approach. In particular, the potential performance degradation when using a reduced training set, as well as the possible impact on generalization under domain shifts, are not adequately analyzed.

**Strengths And Weaknesses:**

Strengths：

1.	The writing in the article is clear and easy to follow.

2.	The experiments and visualizations are comprehensive and well done.

3.	The proposed method outperforms existing CS models on three person ReID datasets and one vehicle ReID dataset.

Weaknesses:

1.	The motivation of applying coreset selection to ReID tasks is somewhat unclear.Compared to other domains such as large-scale language or diffusion models, the training cost in ReID is relatively moderate, and the primary goal is to maximize test-time retrieval performance (e.g., Rank-1, mAP).In this context, reducing the training set size while sacrificing performance raises concerns about the practical utility of the proposed method.

2.	Although the proposed method outperforms other coreset selection approaches in Table 1, the performance gap compared to training on the full dataset remains substantial. Particularly, on MSMT17-V2, the mAP drops from 60.6 to 29.3 and Rank-1 from 81.2 to 52.0, which is a significant degradation. Such a large performance drop raises concerns about whether the reduction in training data is worth the loss in accuracy. These results also suggest that the method struggles on more challenging, large-scale datasets, raising concerns about its robustness and generalization ability.

3.	The paper lacks evaluation on cross-dataset generalization, which is a crucial aspect in ReID. As reducing the dataset may discard informative samples that are critical for handling domain shifts, it is important to assess whether the selected coreset preserves the model's generalization ability.

---

> ### Author Rebuttal · Authors · 2026-03-28
>
> **A1. Motivation of coreset selection for ReID**
>
> In practical real-world applications, ReID data are usually obtained from multi-camera video streams and consist of a large number of image frames with subtle differences, which results in high temporal redundancy. Therefore, this leads to increased storage and training costs when storing and processing such data. Current representative benchmark datasets are curated versions of such massive raw data; for example, the Market1501 dataset is a curated version of the MARS dataset composed of over 1M images [R1]. We can use the proposed CSOR as an essential preprocessing tool to distill a highly informative, non-redundant subset from massive raw data at various scales depending on the application.
>
> Furthermore, as detailed in Appendix A, CSOR can be used for practical resource-constrained applications such as Continual Learning and Neural Architecture Search, which can be used for edge devices (i.e., Jetson Nano [R2] that has only 16GB storage). As demonstrated in the literature [R3–R5], Continual ReID is one of the most practical applications, where systems must adapt to new domains while strictly limiting memory buffers to prevent catastrophic forgetting—aligning with your insight on generalization. In this scenario, CSOR efficiently constructs an optimal exemplar set for continual learning under a strict budget and outperforms baselines in preserving past knowledge, as shown in Fig. 9.
>
> We will clarify the motivation of our research by integrating practical applications into the main text.
>
> [R1] "MARS: A video benchmark for large-scale person re-identification." ECCV. 2016.
> [R2] https://developer.nvidia.com/embedded/jetson-nano
> [R3] "Lifelong person re-identification by pseudo task knowledge preservation." AAAI. 2022.
> [R4] "LSTKC: Long short-term knowledge consolidation for lifelong person re-identification." AAAI. 2024.
> [R5] "Diverse representations embedding for lifelong person re-Identification." IEEE TNNLS. 2025.
>
> **A2. Effectiveness and robustness of the proposed method**
>
> Thanks for this insightful comment. It is worth noting that, while intra-class diversity is a primary requirement for coresets in ReID, the proposed Coverage-Prioritized Sampling (CPS) adaptively selects samples according to class complexity until sufficient intra-class diversity is achieved, thereby allocating more budget to more complex classes.
>
> In this paper, we reported experimental results under fixed budget settings, as practical hardware constraints (e.g., edge devices) impose a fixed memory capacity. We would like to clarify that the performance drop on MSMT17-V2 primarily arises from the stringent budget of 1,500 images—accounting for only 4.6% of the entire dataset—rather than dataset-dependent behavior.
>
> To verify the scalability on MSMT17-V2, we report performance (mAP / Rank-1) under progressively larger budgets:
>
> | Budget (Ratio)| Forgetting | EL2N | AUM | Proposed |
> |:---:|:---:|:---:|:---:|:---:|
> | **3,000 (9.2)** | 5.6 / 14.4 | 15.2 / 33.2 | 25.2 / 49.5 | **37.4 / 61.5** |
> | **5,000 (15.3)**| 11.5 / 25.1 | 30.5 / 55.4 | 31.0 / 56.3 | **42.3 / 66.6** |
> | **7,000 (21.5)**| 28.2 / 50.9 | 37.8 / 62.9 | 35.4 / 60.1 | **45.1 / 69.2** |
> |**16,311 (50.0)**| 45.4 / 68.9 | 49.8 / 73.5 | 50.2 / 73.5 | **55.6 / 77.7** |
>
> As the budget increases, the performance gap relative to full-dataset training diminishes. In particular, when the budget is scaled to 50% of the dataset, our method achieves 91.7% of the performance obtained with full-dataset training. Moreover, our method quickly recovers full-dataset performance and consistently outperforms all baselines.
>
> **A3. Cross-dataset generalization**
>
> To evaluate the generalization ability of the proposed method, we conducted cross-dataset experiments under a 1,000-image budget.
>
> | Source $\rightarrow$ Target | Forgetting | EL2N | AUM | Proposed | Whole |
> | :---: | :---: | :---: | :---: | :---: | :---: |
> | **Market $\rightarrow$ CUHK** | 5.4/4.2 | 13.5/12.5 | 13.6/12.0 | **17.4/16.3** | 21.9/21.4 |
> | **Market $\rightarrow$ MSMT** | 6.7/20.6 | 10.3/28.4 | 11.4/29.8 | **11.5/30.1** | 11.9/30.8 |
> | **MSMT $\rightarrow$ CUHK** | 1.8/1.2 | 1.9/0.9 | 8.3/7.9 | **15.0/13.5** | 24.4/22.9 |
> | **MSMT $\rightarrow$ Market** | 8.3/21.6 | 10.1/26.2 | 27.1/51.1 | **35.3/59.5** | 39.3/65.8 |
>
> Rather than approximating global distributions, our method minimizes redundancy while preserving intra-class diversity, thereby encouraging domain-agnostic features and mitigating overfitting. Notably, CSOR outperforms all compared methods and, in some cases, achieves performance comparable to whole-dataset training. This observation suggests that training on the full dataset can introduce source-specific distributional biases that may impair generalization.
> We will add these results to the revised manuscript.

---

> > ### Author Rebuttal · Reviewer_vf2c · 2026-04-03
> >
> > Thank you for your thorough responses. I have no further concerns, and I will adjust my score accordingly.

---

> > > ### Author Response · Authors · 2026-04-03
> > >
> > > We are pleased that your concerns have been fully addressed. We sincerely appreciate your constructive feedback and your consideration in raising the score.

---

### Decision · Program_Chairs · 2026-04-30

**Decision:**

Accept (regular)

**Comment:**

The paper received all positive recommendations. After rebuttal, all reviewers acknowledged the strengths of the paper and agreed that the authors have addressed their concerns. Reviewers achieved a final consensus of positive rating of this paper. AC agrees with this recommendation and therefore is happy to accept the paper. Authors are required to update the rebuttal and discussion contents to the camera-ready version of the paper to improve it so as to address the raised concerns in the final paper.